# Active diffusion in oocytes nonspecifically centers large objects during prophase I and meiosis I

Alexandra Colin[1] , Gaëlle Letort[2] , Nitzan Razin[3] , Maria Almonacid[2] , Wylie Ahmed[4], Timo Betz[5] , Marie-Emilie Terret[2] , Nir S. Gov[3],
Raphaël Voituriez[6], Zoher Gueroui[1], and Marie-Hélène Verlhac[2]

**Nucleus centering in mouse oocytes results from a gradient of actin-positive vesicle activity and is essential for developmental success. Here, we analyze 3D model simulations to demonstrate how a gradient in the persistence of actin-positive vesicles can center objects of different sizes. We test model predictions by tracking the transport of exogenous passive tracers. The gradient of activity induces a centering force, akin to an effective pressure gradient, leading to the centering of oil droplets with velocities comparable to nuclear ones. Simulations and experimental measurements show that passive particles subjected to the gradient exhibit biased diffusion toward the center. Strikingly, we observe that the centering mechanism is maintained in meiosis I despite chromosome movement in the opposite direction; thus, it can counteract a process that specifically off-centers the spindle. In conclusion, our findings reconcile how common molecular players can participate in the two opposing functions of chromosome centering versus off-centering.**

## Introduction

The position of the nucleus in a cell can instruct morphogenesis, conveying spatial and temporal information. In addition, abnormal nuclear positioning can lead to disease (Gundersen and Worman, 2013). In mammals, the oocyte nucleus is centered via actin-based mechanisms (Almonacid et al., 2015). Importantly, an off-centered nucleus correlates with poor outcomes for mouse and human oocyte development (Brunet and Maro, 2007; Levi et al., 2013). While centering of the nucleus might seem surprising in oocytes subsequently undergoing two extremely asymmetric divisions in terms of the size of the daughter cells, requiring an off-centering of their chromosomes (Azoury et al., 2008; Dumont et al., 2007; Verlhac et al., 2000a), we recently showed that nucleus centering in mouse oocytes modulates gene expression (Almonacid et al., 2019).

We discovered how the nucleus is actively centered in mouse oocytes (Almonacid et al., 2015). We observed that oocytes derived from Formin 2 knockout (*Fmn2−/−*) mice, which lack actin filaments in their cytoplasm, present off-centered nuclei (Azoury et al., 2011; Dumont et al., 2007). Formin 2 is a straight microfilament nucleator and also an essential maternal gene (Leader et al., 2002). Remarkably, the reintroduction of Formin 2 into *Fmn2−/−* oocytes, which harbor initially off-centered nuclei, induces the nucleation of a cytoplasmic actin mesh and a

directional motion of the nucleus toward the center within ∼5 h (Almonacid et al., 2015). We found evidence suggesting the existence of a centering force exerted by the actin mesh, akin to an effective pressure gradient, that acts on the nucleus to move it from the periphery to the center (Almonacid et al., 2015). In the mouse oocyte model system, actin filaments are nucleated from Rab11a-positive vesicles by two types of actin nucleators, Formin 2 and Spire 1&2, which are anchored on these vesicles (Schuh, 2011). We showed that the activity of these actin-positive vesicles decreases from the cortex to the oocyte center as quantified by their squared velocity. On the basis of a simple model describing the pool of actin-positive vesicles as an ideal suspension of self-propelled particles, we proposed that this gradient of activity of actin vesicles, which move by active diffusion (Almonacid et al., 2015), generates an effective pressure gradient (Razin et al., 2017b,a; Solon et al., 2015) and thus a propulsion force. It would therefore be the driver of nuclear motion toward the oocyte center (Almonacid et al., 2018). Interestingly, recent evidence has shown that active diffusion is also a major player in organelle motion in the cytoplasm of *Drosophila melanogaster* oocytes (Drechsler et al., 2017).

In previous work, we used analytical modeling to show that, in principle, a gradient of active particles can center objects

.............................................................................................................................................................................................................................
[1]Department of Chemistry, Ecole Normale Supérieure, Paris Sciences et Lettres Research University, CNRS-ENS-UPMC 24, Paris, France;   [2]Center for Interdisciplinary Research in Biology, Collège de France, CNRS, INSERM, Paris Sciences et Lettres Research University, Equipe Labellisée Fondation pour la Recherche Médicale, Paris, France;   [3]Department of Chemical and Biological Physics, Weizmann Institute of Science, Rehovot, Israel;   [4]Department of Physics, California State University, Fullerton, CA;   [5]Institute of Cell Biology, Cells in Motion Interfaculty Center, Centre for Molecular Biology of Inflammation, Münster, Germany;   [6]UMR 8237 and UMR7600-CNRS/Sorbonne Université, Paris, France.

Correspondence to Marie-Hélène Verlhac: marie-helene.verlhac@college-de-france.fr;   Zoher Gueroui: zoher.gueroui@ens.fr.

(Razin et al., 2017b,a). Here, we used 3D numerical simulations to allow direct comparison between the model and the experimental data. This allowed us to demonstrate that a gradient of persistence of actin-positive vesicles indeed recapitulates many observed features of nucleus centering in the oocyte, using parameters extracted from experiments (Almonacid et al., 2015). Our analytical modeling and 3D simulations suggest that the active pressure-centering mechanism should not be specific to the nucleus.

We tested this by microinjecting oil droplets as well as fluorescently labeled latex beads of various sizes and by monitoring their dynamics. This allowed us to probe the spatiotemporal rheological properties of the actin cytoskeleton and analyze the transport properties of exogenous passive particles of different sizes and chemical nature. Nuclear-sized but fully passive oil droplets were centered with velocities comparable to those of the nuclear ones. This indicates that the centering mechanism is nonspecific and does not require any specific signaling to move the oocyte nucleus. These results support the proposed pressure gradient mechanism, which has exactly these properties and is able to center other objects in addition to the oocyte nucleus. From our simulations, we predict that there is a critical size threshold, whereby objects below a few micrometers should not be sensitive to the gradient of pressure. Consistently, our experiments show that objects larger than a few micrometers in diameter experience a biased movement toward the center of the oocyte.

In addition, a puzzling question in the field is how the same molecules, namely, Formin 2, Spire 1&2, and Myosin-Vb, are able to promote two opposite motions: centering of chromosomes in prophase I (Almonacid et al., 2015) and off-centering of chromosomes later in meiosis I (Azoury et al., 2008; Chaigne et al., 2013; Holubcová et al., 2013; Pfender et al., 2011; Schuh, 2011; Schuh and Ellenberg, 2008). Our results offer an explanation to this long-standing question. Indeed, oil droplets injected in meiosis I also undergo centering in a cytoplasm that is mechanically comparable to that of prophase I oocytes as measured by optical tweezers. Thus, a similar nonspecific mechanism to center organelles coexists in meiosis I together with a specific process that depends on another motor, Myosin-II, which promotes spindle off-centering (Chaigne et al., 2013, 2015; Pfender et al., 2011; Schuh and Ellenberg, 2008).

## Results

### 3D simulations of the centering mechanism driven by a gradient of activity

In previous work, we proposed physical models to describe how centering of the nucleus could be due to a gradient of actin-positive vesicle activity (Almonacid et al., 2015; Razin et al., 2017b,a). In these theoretical descriptions, the observed gradient of activity (Almonacid et al., 2015) leads to a nonspecific effective pressure gradient toward the oocyte center acting on any particle immersed in the cytoplasm. This active pressure can be treated in a simplified manner as a collection of self-propelled particles (actin-positive vesicles) that have a distinct propulsion force, giving them an intrinsic velocity $v$ and a persistence time

$\tau$, decreasing with the distance from the cell cortex. Within such models, because the persistence length $l_p = v\tau$ of the vesicle motion is small compared with that of other length scales in the system (Razin et al., 2017a), the force with which the vesicles push an object, such as the nucleus, is predicted to be proportional to the gradient in persistence length $\nabla l_p$, directed toward decreasing values of $l_p$.

To further test the plausibility of these descriptions in oocytes, we developed 3D agent-based simulations of a system composed of many active particles (the actin-positive vesicles; in red on Fig. 1 A; Romanczuk et al., 2012) and a single passive one (the nucleus; in blue on Fig. 1 A). Given the geometry of the particles, we represented them as individual spheres (off-lattice center-based model; Camley and Rappel, 2017; Van Liedekerke et al., 2015): Each particle was described by the coordinates of its centroid and by its radius (see Materials and methods and Table S1). The motion of each particle was dampened according to Stokes' law; thus, the effect of friction increased linearly with the particle radius. Active particles were set to follow a persistent random motion (Selmeczi et al., 2005), with velocities similar to the propulsion provided by Myosin-Vb (Almonacid et al., 2015; Schuh, 2011). All particles were confined within a sphere of 70-µm diameter, mimicking the oocyte volume (oocyte volume in gray on Fig. 1 A). Based on experimental measurements (Almonacid et al., 2015; Holubcová et al., 2013; Schuh, 2011), we simulated a total of 500 actin-positive vesicles of 1-µm diameter on average and a nucleus of 25-µm diameter. To generate a gradient of activity, we tested the scenario whereby actin-positive vesicles displayed a gradient in the persistence of their motion (how long they maintained a certain directionality), such that persistence was highest at the cortex and decreased toward the oocyte center. We calibrated the motility properties of the simulated vesicles (intrinsic velocity, parameters of linear gradient of persistence) directly with experiments to match the profile of their squared velocities as a function of the distance (Almonacid et al., 2015). With a minimal persistence time $\tau_0$ of 0.001 min and a slope of the gradient in the persistence time $\tau_r$ of 0.25 min·µm$^{-1}$ (see Material and methods), the squared velocity of simulated vesicles displayed a profile that was indeed similar to the measured one (Fig. 1 B, compare the distribution of red dots for simulations to the black line for experimental data and also compare the distribution of simulations versus the display of all experimental data presented in Fig. S1 A).

Using these values in the simulation, the increased persistence of active particles near the cortex resulted in a progressive push of the nucleus toward the oocyte center (Fig. 1, A and C; and Video 1). The resulting motion of the nucleus obtained from the simulations was directional (Fig. 1 D), as observed experimentally (Almonacid et al., 2015). In all simulations, the nucleus was centered after 1,000 min. It reached the center of the oocyte within 400 min on average, taking slightly longer than in experiments (around 300 min; Almonacid et al., 2015) but with a radial velocity profile comparable to that of the experimental one (Fig. 1 E, compare the blue curve and its variations for simulations to the dark triangles for experiments). When it reached the oocyte center,

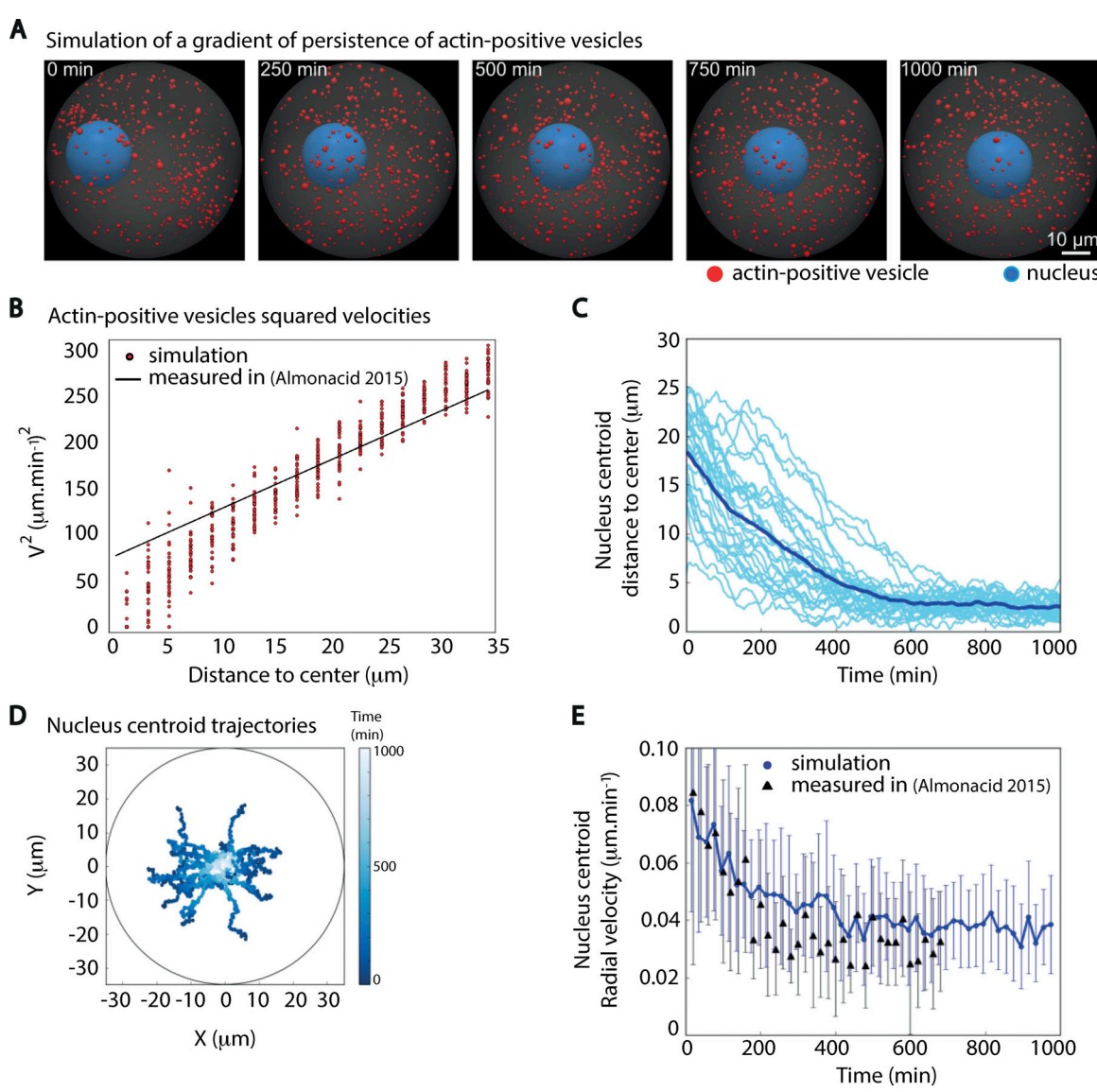

**Figure 1. Simulating the effect of a persistence gradient on the motion of a spherical object similar in size to the nucleus. (A)** Time frames from one 3D simulation of a gradient of persistence of actin-positive vesicles. The images correspond to Video 1. The nucleus is in blue, the actin-positive vesicles are in red, and the oocyte cytoplasm appears in gray. **(B)** Squared velocities of actin-positive vesicles as a function of the distance to the oocyte center during the centering phase. One red dot corresponds to the averaged square velocities of vesicles in the given distance bin in one simulation, taken between 30 and 60 min of simulation only. The black line represents the fit extracted from experimental data in Almonacid et al. (2015). **(C)** Evolution of the distance between the nucleus centroid to the center of the oocyte as a function of time for 30 different simulations. The mean curve appears in dark blue while all other curves appear in lighter blue. **(D)** Trajectories of the nucleus centroid from the cortex to the oocyte center for the 30 simulations in C. The coordinates of the nucleus centroid (X,Y) are given in the oocyte referential where (0,0) is the oocyte center. The running time for the trajectories is presented as a heat map of lighter blue colors (bar on the right side of the picture). The simulations show that the motion of the object representing the nucleus is rather directional to the oocyte center. **(E)** Measure of the radial velocity of the nucleus centroid as a function of time from the 30 simulations confronted to the experimentally measured ones. The radial velocities from the simulations are in blue (dark blue curve for the average curve and lighter blue for the standard deviation at each point of the average curve), while the experimentally measured ones (Almonacid et al., 2015) appear as dark triangles. The experimental values are in the range of the simulated ones.

the nucleus centroid maintained a basal radial velocity in the range of what has been observed experimentally (~0.04 µm·min⁻¹; Almonacid et al., 2015). The remaining basal velocity of the nucleus centroid argues that the nucleus stays dynamically at the center at steady state in a control oocyte, as observed (Almonacid et al., 2015, 2019).

We checked that the centering was truly a consequence of the simulation of a gradient of persistence of actin-positive vesicles by implementing active particles with a homogeneous persistence throughout the cell, but with an average velocity similar to the mean measured one (Fig. S1, B–D). As expected, without any gradient of activity (Fig. S1 D), the nucleus was pushed by active particles but did not display a net directional motion toward the center (Fig. S1, B and C; and Video 2). Importantly, the simulation of a persistence gradient exhibited a spatially uniform density of actin-positive vesicles, as was observed experimentally

(Fig. S1 E; Almonacid et al., 2015) and as expected from theory (Razin et al., 2017a).

Thus, our simulations of a persistence gradient tuned to in vivo–like properties reproduced the centering effect on the nucleus with dynamics similar to the ones observed experimentally. This demonstrated that with the used parameter, the resultant pressure effect was comparable to the observed centering, hence strongly reinforcing the plausibility of a centering mechanism due to a gradient of activity of actin-positive vesicles. The new evidence presented here implies that the centering mechanism is nonspecific and should also apply to inert objects.

### The centering mechanism is not specific to the biological nature of the nucleus

We first used inert oil droplets as passive objects to experimentally test the implication of our theoretical approach, which suggested that the centering mechanism should not be specific to the nucleus. For this, we injected oil droplets at the periphery of prophase I mouse oocytes (Fig. 2 A, yellow arrow) and observed their behavior after injection. These oil droplets present 1.9× the density of water and maintain a round shape, indicating that the forces generated in the cytoplasm are much weaker than the surface tension of the droplet.

We followed the oil droplets in prophase I oocytes over the course of 15 h with a low temporal resolution (Δt = 20 min). Interestingly, they were progressively centered (Fig. 2 A and Video 3) and remained in the proximity of the oocyte center (Fig. S2, A and B). The oil droplet injection technique allowed the production of droplets with diameters in the range of 5 to almost 30 μm that could be optically tracked (see Materials and methods and Fig. S2 C), with some droplets presenting a size comparable to the nucleus (25 μm from Almonacid et al., 2015). All droplets displayed a movement that was directed toward the central area of the oocyte (Fig. 2, B and C), as observed for the nucleus (Almonacid et al., 2015).

To further characterize the motion of oil droplets, we recorded movies at higher temporal resolution (Δt = 500 ms) but on shorter time courses (4 min; Video 4). From these videos, we tracked the oil droplet centroid to analyze the nature of its motion (see Materials and methods). First, we measured the mean squared displacement (MSD; Fig. 2 D) of the oil droplet centroid. The MSD displayed a linear dependence in time on a log/log scale (Fig. S2 D), indicative of a diffusive motion with a fitted diffusion coefficient of 0.072 $\mu m^2 \cdot min^{-1}$ (Fig. S2, E and F). On the contrary, in oocytes treated with cytochalasin D (Cyto D), which induces the depolymerization of actin microfilaments, the droplets were almost immobile (Fig. 2 D, compare black and blue curves; and Video 5).

Then, to check for a potential directional bias in the droplets' motion, detectable on short time scales, we calculated the percentage of displacements toward the oocyte center over the total displacements of the droplet centroid (Fig. S3 A, scheme). This percentage was averaged over all the points of the trajectories for all droplets on a given time step Δt (named hereafter cumulative bias; see Materials and methods). The cumulative bias toward the center was significantly higher than would be observed from purely random trajectories (Fig. S3 B; the values are

outside the confidence interval of random motion in gray; see Materials and methods). On the contrary, the cumulative bias toward the cortex was below the noise and thus indicative of an absence of bias in this direction (Fig. S3 B). In oocytes treated with Cyto D, both types of cumulative bias, toward the center or toward the cortex, were within the range for random trajectories, suggesting an absence of biased motion (Fig. S3 C). Altogether, our experimental data suggest that the motion of oil droplets inside the cytoplasm is biased toward the center of the oocyte when F-actin is present.

Since the dynamics of the cytoplasmic actin mesh is controlled by the Myosin-Vb motor (Holubcová et al., 2013), this observation confirms that oil droplets undergo active (i.e., nonthermal) diffusion, as observed for vesicles (Almonacid et al., 2015). In conclusion, oil droplet movement can be described by a biased diffusion toward the oocyte center, mediated by the activity of the F-actin meshwork (Fig. 2 E).

Together, our results demonstrate that the F-actin–dependent centering mechanism in mouse oocytes in prophase I is general in nature and applies to inert objects.

### The oil droplet recapitulates the F-actin interaction and the movement of the nucleus

When oil droplets were injected in the presence of an F-actin probe (GFP-UtrCH; Burkel et al., 2007), we observed actin filament accumulation around the droplet and the nucleus. The density of the meshwork around the droplet seemed similar to the one surrounding the nucleus and was strictly dependent on the level of expression of the F-actin probe (Fig. 3 A, high expression levels on the left panels and moderate expression levels on the right panels). To quantify the local increase in density of the actin meshwork, the fluorescence intensity of the probe was measured on the droplet or on the nucleus in a condition of high levels of GFP-UtrCH expression and compared with its intensity in the cytoplasm (Materials and methods). This analysis revealed an enrichment of F-actin around the nucleus and the droplet (Fig. 3 B). This enrichment of F-actin is ~50% larger in both cases (droplet and nucleus) compared with the bulk of the cytoplasm (Fig. 3 B). This nonspecific accumulation of actin on the surface of the passive object is expected from the persistent nature of the motion of the actin-positive vesicles (Razin et al., 2017a). However, we do not consider it as having physiological relevance, but rather, it is a side effect due to the presence of boundary conditions favorable to filament nucleation, both in the case of the nuclear envelope and the oil droplet surface (as described in Vignaud et al., 2012). Nevertheless, our observations argue that the oil droplets display properties of local F-actin enrichment that resemble the ones observed for the nuclear envelope.

We then compared the radial velocities (defined as the radial component of the displacement vector over elapsed time) of droplets with a diameter between 20 and 30 μm, in the range of the nucleus diameter (Fig. 3 C), to the radial velocities of the nucleus centroid from $Fmn2^{-/-}$ oocytes reexpressing Formin 2 (Almonacid et al., 2015). The radial velocities were comparable for oil droplets and the nucleus (Fig. 3 C, compare green and red points). In both cases, we observed a decrease in the velocity at

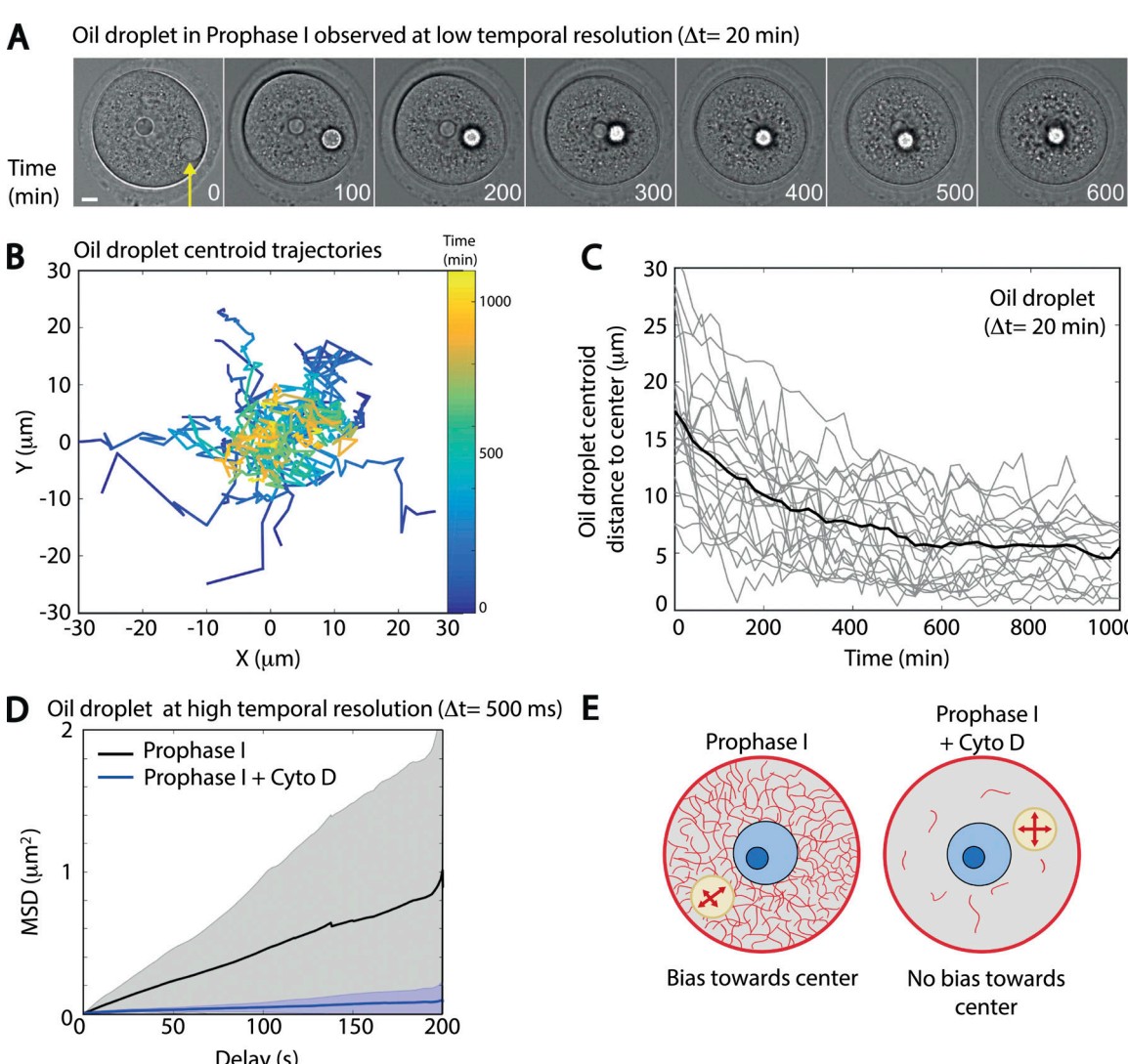

Figure 2. **Peripherally injected oil droplets are centered in oocytes in prophase I. (A)** Transmitted light images of an oil droplet moving toward the center in a prophase I oocyte (observation at low temporal resolution, Δt = 20 min). Images correspond to Video 3. Frame interval displayed here is every 100 min. Scale bar is 10 µm. The yellow arrow points at the oil droplet. The highly refringent object observed by transmitted light on the first frame of the movie corresponds to the nucleolus inside the nucleus. **(B)** Trajectories of centroid droplets. The coordinates of the droplet centroid (X,Y) are given in the oocyte referential where (0,0) is the oocyte center. Time (in minutes) is encoded in color on the droplet centroid trajectories (colored bar on the right side of the picture). n = 22 oocytes; three independent experiments. **(C)** Distance of the droplet centroid to the oocyte center presented as a function of time (in minutes) for each individual droplet (gray curves). The black line represents the distance to the oocyte center as a function of time (in minutes) averaged from the whole dataset of droplet centroid trajectories. n = 22 oocytes; three independent experiments. **(D)** MSD in micrometers squared of the droplet centroid as a function of the delay (in seconds) for droplets injected in prophase I and observed at high temporal resolution (Δt = 500 ms). n = 29 oocytes in prophase I, and n = 14 oocytes in prophase I + Cyto D. Mean MSD curves are in bold (dark black curve for controls; dark blue curve for Cyto D–treated oocytes). The standard deviations of the MSD curves are presented in gray and light blue quadrants; three independent experiments. **(E)** Scheme summarizing the fact that the bias toward the center is detected only in the presence of F-actin. The nucleus is in light blue, the nucleolus in dark blue, and the oil droplet in light brown; microfilaments appear in red.

the center, consistent with a vanishing bias. The radial velocities of centroid droplets have a much larger variability than the one measured for the nucleus. These slight quantitative differences could be due to different effects. First, the diameter of oil droplets is more variable than the nucleus diameter, since it is not genetically encoded but controlled manually (see Materials and methods). Second, in the case of the nuclear repositioning, complementary RNA (cRNA) encoding for Formin 2 was injected into $Fmn2^{-/-}$ oocytes; thus,

nucleus centering is occurring while the F-actin mesh is progressively reforming. In contrast, in the case of oil droplets, these are injected at steady state in control oocytes that already present a fully dynamic actin mesh. Third, the droplet is made of incompressible oil and has different mechanical properties than a nucleus has.

Altogether, our results argue that oil droplets display F-actin interaction properties as well as a motion inside the cytoplasm reminiscent of the nuclear ones, making them well-suited

Colin et al.
Nonspecific centering of large objects in oocytes

**Journal of Cell Biology** 5 of 15

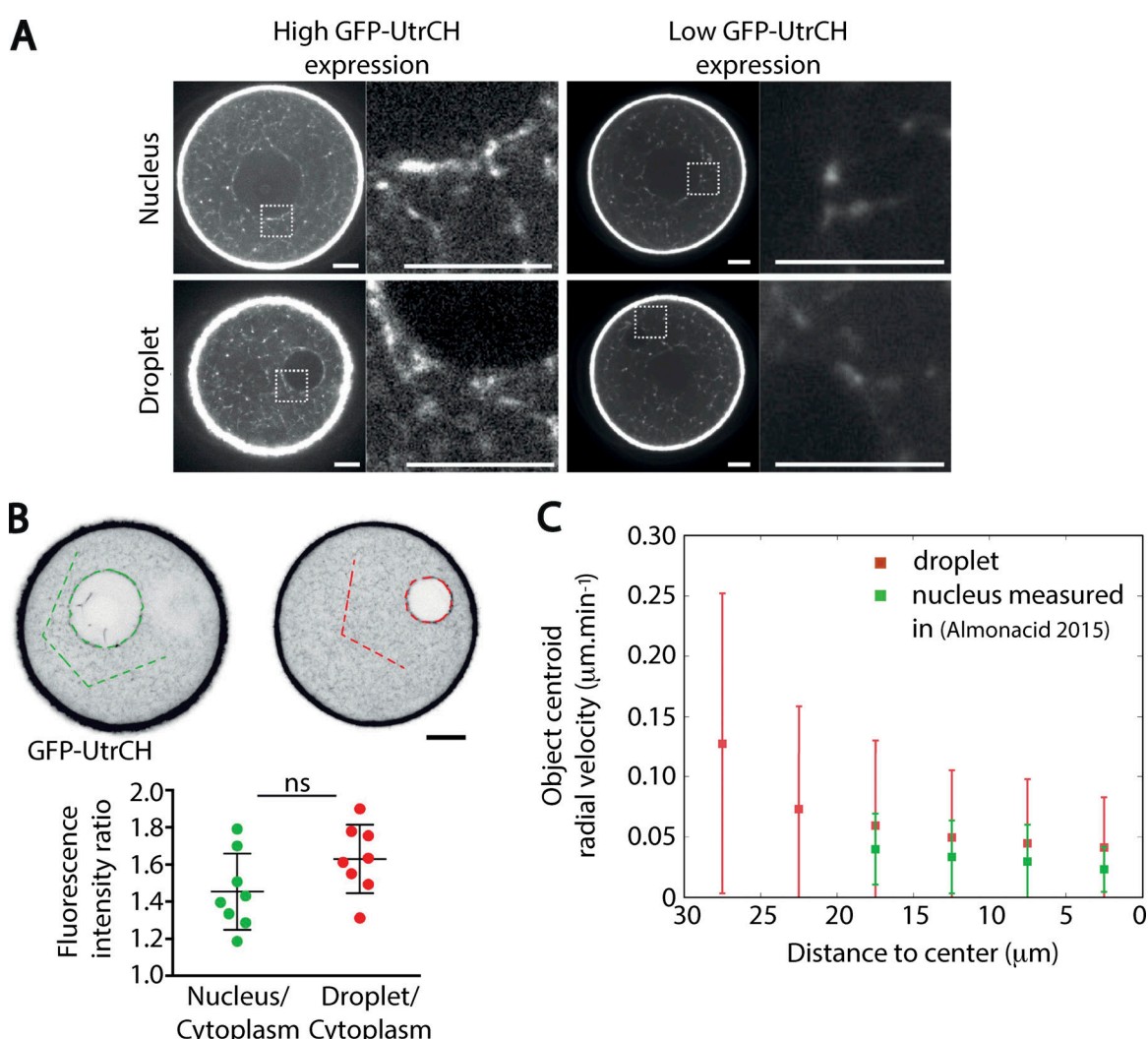

Figure 3. **Oil droplets recapitulate nucleus behavior. (A)** Visualization of F-actin in the region of the nucleus (upper panels) and of the droplet (lower panels) observed in the same oocyte in two different Z planes. F-actin is labeled with GFP-UtrCH, which is either highly expressed (left panels) or mildly expressed (right panels). The square-dashed region is enlarged in the right panel for each condition of GFP-UtrCH expression. F-actin is in white. Scale bars are 10 μm. One experiment. **(B)** Quantification of the ratio of the GFP-UtrCH fluorescence intensity for the nucleus versus the cytoplasm (green dots) and for the oil droplet versus the cytoplasm (red dots) from A. F-actin is in black. The dotted green line and red line overlays on the fluorescent images depict how the different fluorescence intensities were measured (see also Materials and methods). A Kolmogorov-Smirnov test resulted in a nonsignificant (ns) difference between the two distributions (P = 0.28). Scale bar is 10 μm. **(C)** The distributions of the radial velocities of the nucleus centroid and of the oil droplet centroid as a function of the distance to oocyte center are comparable. Red: Radial velocity of oil droplet centroid for droplets with a diameter between 20 and 30 μm. n = 6 oocytes. Green: Radial velocity of nucleus centroid for *Fmn2^{-/-}* oocytes injected with Formin 2 from Almonacid et al. (2015). Mean and standard deviation are superimposed; three independent experiments.

experimental tools to investigate the impact of the actin mesh on cytoplasmic objects.

### Small objects are not subjected to the centering force

To further test our pressure gradient model, we explored the impact of the size of objects on their centering efficiency. First, we tested the size impact in our 3D numerical simulations. For this, we varied the size of the nucleus-like object, ranging from 1 to 36 μm in diameter. We observed an increasing efficiency of centering for larger objects (Fig. 4 A). While the outcome of the simulations indicated a clear correlation between object size and centering efficiency (Fig. 4 C, multicolor points fitted with a regression depicted as a blue dotted line), our experimental

points coming from oil droplets that we could follow throughout the whole movie duration up to their centering, with diameters between 8 and 26 μm, showed only a modest trend (Fig. 4 C, black triangles). This could come from the low number of experimental data (15) or from the limited range of sizes that we could follow on a long time scale with the oil droplet technique. Nevertheless, more efficient centering observed for larger objects is expected from the Archimedes-like property of the centering force (Razin et al., 2017a). Interestingly, this was observed previously for multicellular clusters in a different model system, the *Drosophila* embryos (Cai et al., 2016).

In the simulations, small objects (between 1 and 6 μm in diameter) were not centered, despite a very long simulation

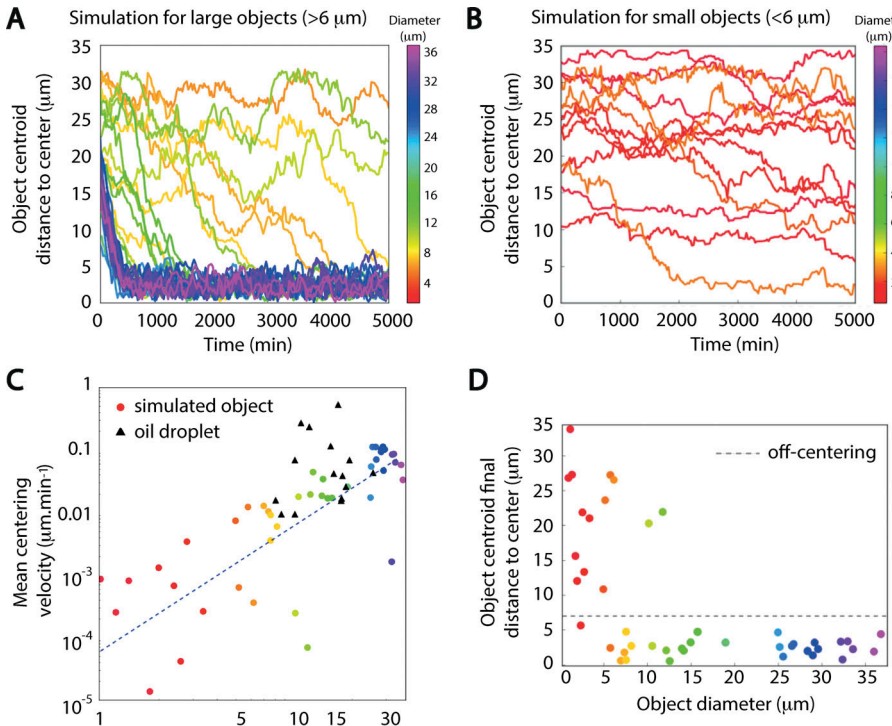

**Figure 4. Prediction from simulations of object behavior as a function of object size. (A)** Evolution of the distance of the nucleus centroid to the oocyte center in 30 different simulations for objects with a diameter >6 µm. One line corresponds to one simulation, and the color indicates the value of the diameter for each object (heat map of diameter sizes is presented on the right scale bar). Most objects are centered before 5,000 min. **(B)** Evolution of the distance of the nucleus centroid to the oocyte center in 15 different simulations for objects with a diameter <6 µm. One line corresponds to one simulation, and the color indicates the value of the diameter for each object (heat map of diameter sizes is presented on the right scale bar). The small objects are not centered even after 5,000-min-long simulations. **(C)** Nucleus centroid mean centering velocity (as defined in Fig. 6 B) in the 45 simulations from A and B as a function of object diameter (colored circles) and mean centering velocity of the oil droplets in our experiments as a function of their diameter (black triangles). The fitted line (dotted blue line) follows the expected relation $v \sim R^2$ from the Archimedes principle for the simulated data ($v = 0.00024\ R^2$, Spearman correlation of 0.8 with $P = 10^{-11}$). **(D)** Final distance of the object centroid to the oocyte center, as a function of the object diameter for the 45 simulations. Objects ending up within 7 µm of the oocyte center (dashed gray line) at the end of the simulation are considered centered (as defined in Fig. 6 B).

occurrence corresponding to an 83-h duration (Fig. 4, B and D). Indeed, due to their small surface, collisions with the active particles were too rare to induce a directed motion. Our simulations suggested the existence of a size threshold of ~6 µm for centering (Fig. 4 D).

We further tested experimentally the existence of a size threshold. To do so, not only did we use the oil droplets to produce large objects (between 5 and 30 µm in diameter), but we also injected 100 nm of fluorescent particles; when injected at a high concentration (Materials and methods), they aggregated in the cytoplasm of the oocyte (Fig. S4 A and Video 6). We took advantage of this aggregation, which allowed the production of objects with sizes between 100 nm and 2 µm (measured by the fluorescence intensity of the detected spot; see Materials and methods), as the injection of larger particles was technically challenging.

To check for a potential bias in particle motion, we calculated the particle displacement cumulative bias, as we did for the oil droplets (Fig. S3 A; see Materials and methods), and compared it to the cumulative bias generated by random trajectories of similar characteristics (e.g., size of objects, diffusion coefficient, and number of trajectories analyzed). For particles with diameters <1.5 µm, both the cumulative biases toward the center and toward the cortex of the oocyte were undistinguishable from the ones of random trajectories (Fig. S4 B). Due to technical limitations, particles of larger sizes were extremely rare to obtain (n = 13 compared with few hundreds for other categories of sizes; see figure legends); thus, the comparison of their cumulative

bias with random trajectories was not conclusive (Fig. S4 B, far right panel). Nonetheless, to further explore the possibility of a directional bias, we measured the radial component (d.cos(θ); Fig. S4 C, scheme) of the particle displacement vector over five steps of time (5Δt, 2.5 s). From the distribution of d.cos(θ) for a delay of 2.5 s, we computed the mean instantaneous radial velocity, which was obtained with the following formula:

$$\text{Mean instantaneous velocity} = \frac{\mu}{\Delta t},$$

where, for example, as schematized on Fig. S4 C, µ corresponds to (d1.cos[θ1] + d2.cos[θ2] + d3.cos[θ3] + d4.cos[θ4]).

We found that the mean instantaneous radial velocity increases with the size of the object (Fig. S4 D). A positive mean velocity can be indicative of a biased movement toward the center. For each size of objects, we calculated the probability that the distribution was significantly different from a normal distribution with the same standard deviation and centered in 0 (result of a z-test). Objects >1-µm diameter had a positive mean instantaneous velocity. Moreover, in the case of objects >1.5 µm, the mean instantaneous velocity was higher than that of the droplet, suggesting an effective bias. Importantly, treatment with Cyto D served as a negative control and showed that in the absence of actin filaments, oil droplets do not present a significant bias on the radial component of their displacement (Fig. S4 D). Combined with our previous oil droplet analysis (Fig. S3), these observations revealed that the motion of small objects (<1 µm diameter) is not biased toward the center, while

larger objects (>5 µm) undergo biased motion. The cutoff size for the centering mechanism could not be precisely determined from our short time scale analysis but lies between 1 and 5 µm.

Thus, the existence of a size threshold for centering was also confirmed experimentally. However, the numerical threshold size (6 µm) was higher than the experimental one (between 1 and 5 µm). The centering efficiency (Fig. 4 C) depends on the frequency of collisions and the crowding of the oocyte cytoplasm, which we did not consider in the simulations, yet it might also impact this threshold. Furthermore, our experimental setup (oil droplets with sizes ranging from 5 to 30 µm) was not adapted to probe specifically this 6-µm size threshold determined by the simulations on long time scales (Fig. 4 D). We can nonetheless conclude that our simulations coupled to experimental evidence support the existence of a nonspecific centering mechanism, experienced by objects larger than a few micrometers.

## Oil droplets are centered in oocytes that undergo meiosis I

As introduced previously, an unresolved question in the field is how the same molecules (Formin 2, Spire 1&2, and Myosin-Vb) promote two opposite motions: centering of chromosomes in prophase I (Almonacid et al., 2015) and off-centering of chromosomes later in meiosis I (Azoury et al., 2008; Chaigne et al., 2013; Holubcová et al., 2013; Pfender et al., 2011; Schuh, 2011; Schuh and Ellenberg, 2008). We thus decided to address whether the centering mechanism was maintained during meiosis I. To this aim, we injected oil droplets in prophase I oocytes and then allowed oocytes to synchronously resume meiosis (indicated by nuclear envelope breakdown [NEBD]) and proceed into meiosis I. Meiosis I is a long process taking ~10 h and is described here in minutes after NEBD (Fig. 5 A). First, we observed that oocytes could undergo meiosis I unperturbed, since they succeeded in dividing on time and extruding a first polar body, arguing that the oil droplet is not toxic for the development of the oocyte (Fig. 5 A). It also showed that the oil droplet could coexist in the cytoplasm with the mechanism that promotes off-centering of the first meiotic spindle, a prerequisite for the first asymmetric division (i.e., first polar body extrusion; Verlhac et al., 2000a). Second, we observed that the oil droplet was centered during the process of meiosis I (Fig. 5 A and Video 7). Effectively, 92% (11 of 12) of the droplets were centered before the extrusion of the first polar body (Fig. 5, C and E). When the actin network was dismantled (in the presence of Cyto D), no movement of the droplet was observed, arguing that droplet centering is here also a consequence of the presence of actin (Fig. 5, B, D, and F; and Video 8).

These results mean that during meiosis I, when the spindle is migrating toward the cortex, a centering mechanism is nonetheless present. Surprisingly, common molecular actors are at play, such as Myosin-Vb, which is essential for both nucleus centering (Almonacid et al., 2015) and spindle migration (Holubcová et al., 2013). These results show the coexistence of two mechanisms in meiosis I: a specific one that allows off-centering of the spindle toward the cortex, involving Myosin-II activity (Chaigne et al., 2013, 2015; Holubcová et al., 2013; Pfender et al., 2011; Schuh, 2011; Schuh and Ellenberg, 2008),

and a nonspecific one that ensures the centering of big objects on long time scales, which, at least in prophase I, does not require Myosin-II activity (Almonacid et al., 2015). To have a better understanding of this process, we compared the characteristics of droplet centering in prophase I and in meiosis I.

## Comparison of droplets centering in prophase I versus meiosis I

To compare the process of droplet centering between the two stages, we first analyzed the radial velocity of droplets observed in prophase I or in meiosis I (Fig. 6 A). Droplets injected in oocytes undergoing meiosis I presented a slower radial velocity than the ones injected in oocytes in prophase I. To have a quantitative characterization of the centering process, we measured the mean centering velocity (Material and methods; Fig. 6 B, upper panel), with t = 0 corresponding to the beginning of the movie for prophase I and t = 0 corresponding to NEBD for meiosis I. We observed that the mean centering velocity was 4× smaller in meiosis I than in prophase I (0.10 µm·min$^{-1}$ in prophase I compared with 0.024 µm·min$^{-1}$ in meiosis I; Fig. 6 B, lower panel). We checked if we were able to link these data with the rheological properties of the cytoplasm. We performed active micro-rheology experiments using optical tweezers on endogenous vesicles to probe the mechanical response of the oocyte cytoplasm in prophase I and meiosis I (Fig. S5). We found no significant difference in the elastic modulus (G′) between oocytes observed in prophase I and in meiosis I (Fig. S5, left panel). As for the viscous modulus (G′′), a difference between prophase I and meiosis I oocytes was observed (slightly higher in prophase I; Fig. S5, right panel). Note that we obtained a viscosity of the order of 1 Pa.s for the cytoplasm of oocytes maintained in prophase I, similar to what was previously found with magnetic tweezers measurements (Hosu et al., 2008). This analysis suggests that oocytes in prophase I or meiosis I have comparable mechanical properties, which could not explain the observed differences in the centering speed.

The differences in centering could be due to the fact that we performed experiments at steady state in prophase I, while in meiosis I, the mesh was progressively reforming while droplets were followed (Azoury et al., 2011). Alternatively, the number and size of actin-positive vesicles has been shown to vary during oocyte meiosis I (Almonacid et al., 2015; Holubcová et al., 2013; Schuh, 2011). In particular, the number of actin-positive vesicles decreased from 500 to 200 just after NEBD, while their volume tripled. If the centering mechanism is a consequence of the actin-positive vesicle activity as we assumed, this variation should affect the centering efficiency and could potentially explain this decrease. Indeed, by simulating fewer and larger active vesicles in our agent-based model, we obtained a slower centering behavior for a nucleus-like object, which took on average 12 h to be centered (Fig. 6, C and D). The comparison of the mean centering velocity between our previous simulations with 500 vesicles (approximately prophase I–like) and those with 200 vesicles (approximately meiosis I–like) confirmed this drastic decrease in efficiency (Fig. 6 E). This observation was consistent with a key role of actin-positive vesicles in the centering mechanism.

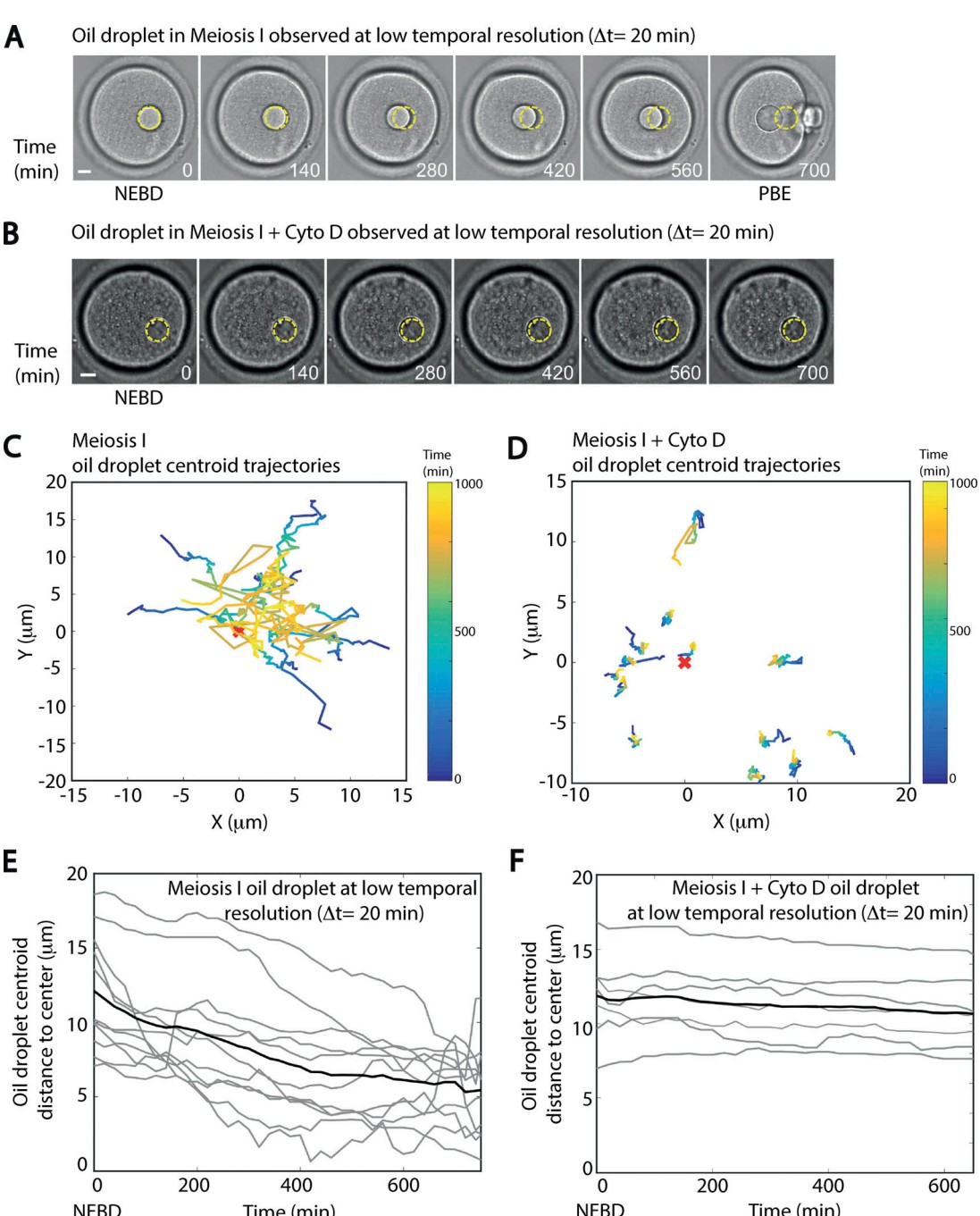

Figure 5. **Oil droplets are also centered in oocytes undergoing meiosis I. (A)** Centering of an oil droplet in an oocyte undergoing meiosis I. The images correspond to Video 7. One frame is shown every 140 min. The first frame corresponds to NEBD, a marker of meiosis resumption, and the last one to polar body extrusion (PBE), a marker of meiosis I completion. The initial location of the oil droplet is highlighted by a dotted yellow circle on each picture. Scale bar is 10 μm. **(B)** An oil droplet observed during meiosis I in an oocyte treated with Cyto D. The images correspond to Video 8. One frame is shown every 140 min. The first frame corresponds to NEBD. Note that oocytes treated with Cyto D do not extrude a polar body. The initial location of the oil droplet is highlighted by a dotted yellow circle on each picture. Scale bar is 10 μm. **(C)** Trajectories of droplet centroids that are centered during the observation in meiosis I. Time (in minutes) is encoded in color on the centroid trajectories (time is presented as a heat map on the right side of the picture). The coordinates of the centroid droplet are given in the oocyte referential where (0,0) is the center of the oocyte. n = 11 oocytes; three independent experiments. **(D)** Trajectories of droplet centroids not centrally located at the beginning of the observation in meiosis I oocytes treated with Cyto D. Time (in minutes) is encoded in color on the centroid trajectories (time is presented as a heat map on the right side of the picture). The coordinates of the centroid droplet are given in the oocyte referential where (0,0) is the center of the oocyte. n = 13 oocytes; two independent experiments. **(E)** Distance of the droplet centroid to the oocyte center presented as a function of time (in minutes) for each individual droplet for oocytes in meiosis I (gray curves). The black line represents the distance to the oocyte center as a function of time (in minutes) averaged from the whole dataset of droplet centroid trajectories. n = 11 oocytes; three independent experiments. **(F)** Distance of the droplet centroid to the oocyte center presented as a function of time (in minutes) for each individual droplet for oocytes in meiosis I treated with Cyto D (gray curves). The black line represents the distance to the oocyte center as a function of time (in minutes) averaged from the whole dataset of droplet centroid trajectories. n = 13 oocytes; two independent experiments.

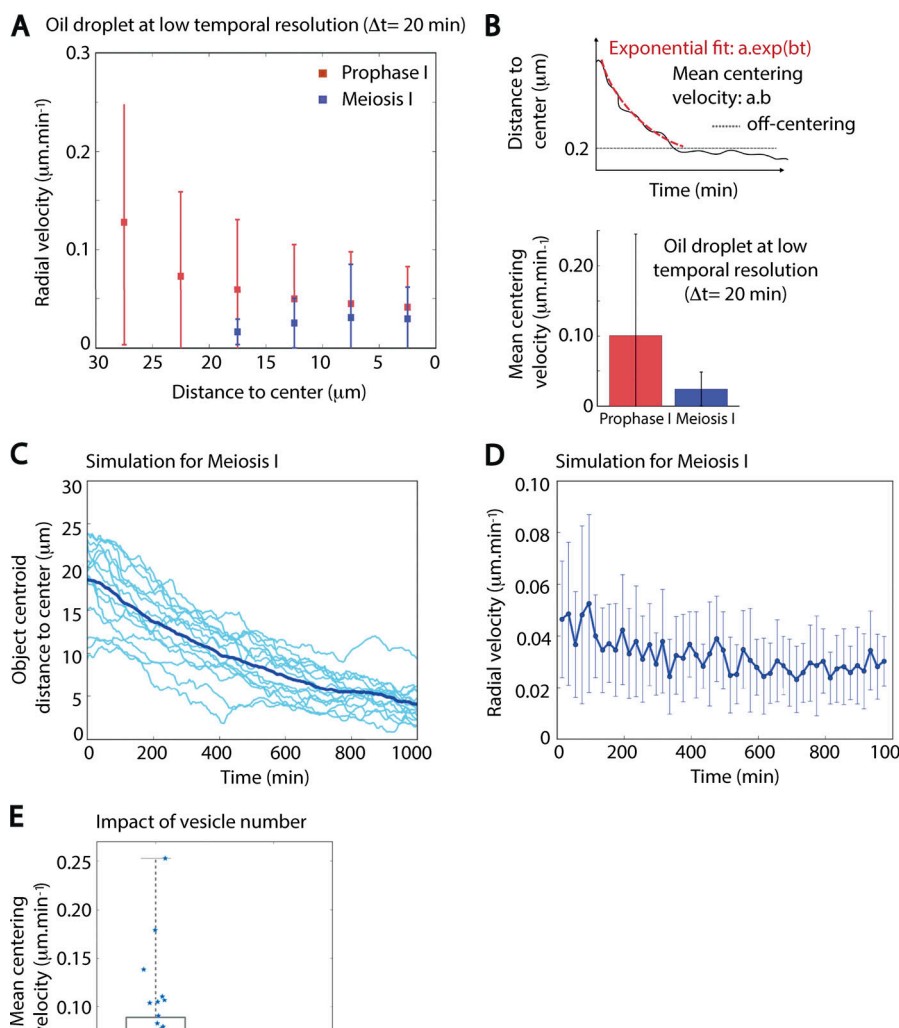

Figure 6. **Experimental and simulated comparison of the efficiency of droplets centering in prophase I versus meiosis I.** **(A)** Radial velocities of droplet centroids as a function of the distance to the oocyte center for droplets centered during prophase I (red squares) and meiosis I (blue squares). Mean and standard deviation are represented. $n$ = 22 oocytes in prophase I and $n$ = 11 oocytes in meiosis I; three independent experiments. **(B)** Top: Scheme explaining the measure of the centering velocity using an exponential fit on the distance to the oocyte center as a function of time. Bottom: Average of the centering velocity (in $\mu m \cdot min^{-1}$) for droplets centered during the experiment in prophase I (red histogram) and meiosis I (blue histogram). $n$ = 22 oocytes for prophase I and $n$ = 11 oocytes for meiosis I; error bars show the standard deviation. Three independent experiments. **(C)** Evolution of the distance of the nucleus centroid to the center of the oocyte as a function of time for 15 different simulations with 200 vesicles per oocyte with a mean radius of 1.4 $\mu m$. The average curve appears in dark blue, while all curves are in lighter blue. **(D)** Measure of radial velocity of the nucleus centroid as a function of time from the 15 different simulations using 200 vesicles per oocyte. The average curve appears in dark blue, while the standard deviation for each point of the average curve appears as lighter blue. **(E)** Impact of the total number of vesicles per oocyte on the mean centering velocity obtained from simulations with $n$ = 500 vesicles (left, 30 simulations) and $n$ = 200 vesicles (right, 15 simulations). 500 vesicles per oocyte corresponds to prophase I, while 200 vesicles per oocyte corresponds to meiosis I.

## Discussion

Using numerical simulations based on a previous analytical model (Razin et al., 2017b,a), we show here that a gradient of persistence of actin-positive vesicles can generate forces driving a nucleus-like object from the periphery to the oocyte center. Experimental injection of oil droplets mimicking the size and F-actin interactions of the nucleus demonstrated that the centering mechanism is indeed nonspecific to the biological nature of the nucleus. In prophase I, oil droplets and nucleus centering have the same overall dynamics. The fundamental reason underlying the persistence of the movement of actin-positive vesicles, observed in Schuh (2011) and confirmed in Almonacid et al. (2015), remains unaddressed. We can only speculate on its origin. It may be related to the presence of a dynamic equilibrium that sustains an important trafficking of Rab11a containing actin-positive vesicles from the center to the periphery (Schuh, 2011). It may also be due to a potential positive feedback loop residing in the massive enrichment of F-actin in and extending from the cortex and/or the accumulation of Myosin-Vb at the cortex (Schuh, 2011).

We tested the influence of object size on centering efficiency. Small objects (below a few micrometers) do not show a biased movement toward the oocyte center at a short time scale (2.5 s), while large objects (above a few micrometers) do. Interestingly, this cutoff is of the same order of magnitude as the estimated meshwork size for oocytes maintained in prophase I (Azoury et al., 2008; Schuh and Ellenberg, 2008) and could explain why particles smaller than the mesh size diffuse freely.

Consistently, our simulations showed that large objects above a few micrometers are centered with a velocity that increases proportionally with their surface. The microinjection of oil droplets displayed a similar tendency. The effectiveness of the centering mechanism can be quantified by the Peclet number, which quantifies whether the bias toward the center is stronger than diffusion for an object. The Peclet number, $P$, is the ratio between the time for diffusion across the system to the time for directed (biased) advection: $P = \tau_D/\tau_A$. The diffusion time is given by $\tau_D = L^2/D$, where $L$ is the system size and $D = D_T + D_A$ is the diffusion coefficient, which arises from both thermal ($T$) and active ($A$) forces. The advection time, due to the net forces acting on an

object, is given by $\tau_A = L/v$, where, in our system, the net advection velocity of the object, $v$, is given by the unbalanced collisions with the actin-positive vesicles that display an activity gradient.

The Peclet number is therefore:

$$P = \frac{\tau_D}{\tau_A} = \frac{Lv}{D}.$$

A large Peclet number means that the diffusion time is long compared with the advection time so that the advection process is dominant. A small Peclet number corresponds to diffusion-dominated dynamics.

Assuming that, despite the contribution of active effects, the diffusion coefficient satisfies the equilibrium scaling such that $D = \sim 1/r$ and that the advection velocity scales as $v = \sim r^2$, as described in Cai et al. (2016), the Peclet number is eventually found to scale as $P = \sim r^3$; it is thus large for large objects and small for small objects. In agreement with the theory, in the simulations of objects with a diameter of 2 µm (6, 12.5, and 18 µm in diameter, respectively), the Peclet number was 0.05 (4, 30, and 140, respectively). The simulations confirmed that for small objects, the Peclet number was <1 so that thermal diffusion dominated advection, while the motion of larger objects was dominated by active diffusion-driven bias. In the experiments, oil droplets presenting a diameter between 8 and 10 µm ($n = 3$) had Peclet numbers between 1 and 2, whereas oil droplets with a diameter >10.5 µm ($n = 12$) had Peclet numbers between 4 and 128. All Peclet numbers for oil droplets were >1, indicative of a regimen dominated by active diffusion-driven bias, consistent with the observed centering of oil droplets. Interestingly, small droplets (with a diameter <10 µm) presented Peclet numbers close to 1, suggesting an increasing contribution of thermal diffusion compared with active diffusion on the object's motion.

Our work provides another example of how F-actin acts as an organizer of the intracellular space by distributing objects in space as a function of their size. For example, in the *Xenopus laevis* oocyte nucleus, F-actin has been shown to act as a stabilizing scaffold that prevents the influence of gravity (Feric and Brangwynne, 2013). It is also extremely relevant in the context of mammalian oocyte physiology, having potential implications on the distribution of organelles in this large cell, where Golgi stacks are known to be micronized (Wassarman and Josefowicz, 1978; Moreno et al., 2002).

We also show that the centering mechanism is still present in meiosis I, with slower kinetics than in prophase I. The slower droplet centering in meiosis I could be explained by the fact that at this stage, the oocyte shares actin resources for two processes: a nonspecific centering mechanism and a specific off-centering of the meiotic spindle via pulling forces exerted by Myosin-II at the spindle poles and through specific F-actin connections between the meiotic spindle poles and the cortex (Azoury et al., 2008; Chaigne et al., 2013, 2015; Holubcová et al., 2013; Pfender et al., 2011; Schuh, 2011; Schuh and Ellenberg, 2008). Sharing of resources for actin networks has been widely studied in fission yeast. This sharing of resources could contribute to the difference in the total number of actin-positive vesicles observed in prophase I (500) versus meiosis I (200; Holubcová et al., 2013; Schuh, 2011). Interestingly, modulating the number of actin-positive vesicles in the simulations

impacts the efficiency of centering-presenting values comparable to experimental ones. Hence, we propose that the difference in efficiency of centering in prophase I versus meiosis I is simply due to a difference in density of the actin meshwork.

It is also interesting to consider the centering mechanism during meiosis I as a mechanism that counteracts spindle migration. Indeed, if spindle migration was dragging all the maternal stores into the polar body, then the asymmetric division would deplete the oocytes from the reserves necessary for future embryo development. It is possible that the centering mechanism in meiosis I is a safeguarding mechanism to preserve most organelles and RNP granules in the oocyte itself instead of them being transported into the polar body.

## Materials and methods
### Numerical simulations
We developed new 3D numerical simulations in C++, using an off-lattice agent-based (center-based) model. Each agent (actin-positive vesicles, nucleus, or oil droplets) was represented as a sphere, thus characterized by its center coordinates and its radius. The spheres were confined inside a spherical boundary, mimicking the oocyte contour. The motion of each agent was determined by the balance of forces it experienced, which corresponds to its intrinsic motility, its contact with other agents, and its contact with the cortex:

$$\vec{v}_i = \frac{1}{\eta_i}\left(\sum_j \vec{F}_r(i,j) + \vec{F}_{ci} + \vec{B}_i\right),$$

with $\eta_i = 6\pi r_i \gamma$ as the friction coefficient opposing the agent motion, calculated according to Stokes' law, where $\gamma$ is the viscosity of the medium. Thus, the friction coefficient increases linearly with the agent radius $r_i$.

Contacts between agents were modeled as a hard-core repulsive force, $F_r$, effective as soon as spheres overlapped (Ghaffarizadeh et al., 2018; Letort et al., 2019):

$$\vec{F}_r = c_r\left(1 - \frac{\left\|\vec{d}\right\|}{d_{eq}}\right)\frac{\vec{d}}{\left\|\vec{d}\right\|},$$

where $\vec{d}$ is the vector between the two sphere centers and $d_{eq}$ is the equilibrium distance (i.e., the sum of the two sphere radii). Thus, when two spheres came into physical contact, the increasing overlap, which generated an increasing deformation of the spheres, created a repulsive force (Drasdo et al., 2007). The strength of the repulsion increased with sphere overlap, accounting for the limited compressibility of the particles.

Similarly, confinement within the oocyte was modeled as a repulsive force, effective as soon as the agent touched the cortex:

$$\vec{F}_{ci} = c_c\left(1 - \frac{\left\|\vec{p}\right\|}{r_i}\right)^2 \frac{\vec{p}}{\left\|\vec{p}\right\|},$$

where $\vec{p}$ is the vector between the agent center and its projection on the cortex surface is the agent radius ($r_i$).

Each agent could move following a persistent random motion $\vec{B}_i$ with a given instantaneous velocity. The direction of the Brownian motion was updated with a probability:

$$prob = \frac{dt}{\tau}$$

at each time step (Ghaffarizadeh et al., 2018) $dt$, thus allowing for a persistence time of the motion controlled by the parameter $\tau$.

To simulate a gradient of persistence, the value of $\tau$ was calculated for each agent at each time step as $\tau_i = \tau_0 \left(1 + \tau_r \left\| \vec{r} \right\| \right)$.

$\tau_0$ and $\tau_r$ are parameters to define the persistence gradient (minimal value and slope), and $\vec{r}$ is the vector between the vesicle center and the oocyte center.

We adapted part of the PhysiBoSS source code (Letort et al., 2019) in our software to handle simulation inputs and outputs, represent them with ParaView software, and analyze them with Python scripts. As in Ghaffarizadeh et al. (2018) and Letort et al. (2019), for running time efficiency, the numerical integration was done with the Adam's Bashforth integration scheme and calculation of the agent motion was parallelized with openMP.

### Oocyte collection, culture, and microinjection
Oocytes were collected from 11-wk-old OF1 mice as previously described (Verlhac et al., 2000b) and maintained arrested in prophase I in M2 + BSA medium supplemented with 1 µM milrinone (Reis et al., 2006). All live culture and imaging was performed under oil at 37°C. We used the following pspe3-GFP-UtrCH (Azoury et al., 2008) construct to produce cRNA. In vitro synthesis of capped cRNAs was performed as previously described (Verlhac et al., 2000a). cRNAs were centrifuged at 4°C during 45 min at 13,000 rpm before microinjection. The cRNA encoding GFP-UtrCH was injected first, and then oocytes were injected with oil droplets. We injected Fluorinert FC-70 (Sigma; Ref. F9880) presenting a density of 1.9× that of water at the maximal pressure of the microinjector (clean mode at 7,000 hPa). The size of the oil droplets was visually adjusted by manual control of the duration of the microinjection pulse. Microinjections were performed using an Eppendorf Femtojet microinjector at 37°C as in Verlhac et al. (2000b).

### Drug treatments
Cyto D (Life Technologies; Ref. PHZ1063) was diluted at 10 mg·ml⁻¹ in DMSO and stored at –20°C. It was used on oocytes at 1 µg·ml⁻¹. For the injection of latex fluorescent beads (0.1 µm; Life Technologies; F8803), the beads were rinsed several times in nuclease-free water before use to remove traces of sodium azide and were diluted 10 times before injection.

### Live imaging
Spinning disk images were acquired at 37°C in M2 + BSA + 1 µM milrinone using a Plan-APO 40×/1.25 NA objective on a Leica DMI6000B microscope enclosed in a thermostatic chamber (Life Imaging Service) equipped with a CoolSnap HQ2/CCD-camera (Princeton Instruments) coupled to a Sutter filter wheel (Roper Scientific) and a Yokogawa CSU-X1-M1 spinning disk. Oil droplet images were acquired using either the stream mode of the camera on Metamorph (one image every 500 ms) or one

image every 20 min to follow the whole motion toward the oocyte center. The actin cytoplasmic meshwork decorated with GFP-UtrCH was imaged using Metamorph upon excitation at 491 nm.

### Optical tweezers experiments
The single-beam gradient force optical trap system uses a near infrared fiber laser (λ = 1,064 nm; YLM-1-1064-LP; IPG) that passes through a pair of acousto-optical modulators (AA-Optoelectronics) to control the intensity and deflection of the trapping beam. The laser was coupled into the beam path via dichroic mirrors (Thorlabs) and focused into the object plane by a water immersion objective (60×, 1.2 NA; Olympus). The condenser was replaced by a long-distance water immersion objective (40×, 0.9 NA; Olympus) to collect the light and imaged by a 1:4 telescope on an InGaAs quadrant photodiode (Hamamatsu; G6849). The resulting signal was amplified by a custom-built amplifier system (Oeffner Electronics) and digitized at a 500-kHz sampling rate and 16 bits using an analogue input card (National Instruments; 6353). The position of the trapped particle was measured by back focal plane interferometry (Gittes and Schmidt, 1998). All control of the experimental hardware was executed using LabVIEW (National Instruments). Optical trapping of endogenous (diameter, ~1 µm) vesicles was calibrated using the active-passive method as in Mas et al. (2013), where the high-frequency fluctuations (f > 300 Hz) are thermal in origin (Ahmed et al., 2018; Fodor et al., 2016). Vesicle size was estimated by comparing image analysis and laser interferometry profiles to 1-µm beads (Ahmed et al., 2018). The mechanical response was measured by applying a sinusoidal force to a vesicle and observing the subsequent displacement. The shear modulus was calculated from the mechanical response using the generalized Stokes-Einstein relation as done previously (Mizuno et al., 2007).

### Image analysis
Image analysis was performed using ImageJ, Icy, and Matlab. When needed, movies were realigned with the StackReg plugin of ImageJ. For all automated tracking, Icy software was used (de Chaumont et al., 2012). Droplet tracking in a bright field at short time scale was done with the Active Contours plugin. Fluorescent particle tracking was done with the Spot Detector plugin combined with the Spot Tracking and Track Manager plugins. The trajectories were exported in Excel files and analyzed using Matlab software. The tracks were filtered to keep only the tracks with >30 points (movies of 400 frames). This threshold was determined with simulated data to avoid the tracking of false trajectories coming from the noise in the movies. MSD analysis was done with the msdanalyzer class in Matlab (Tarantino et al., 2014). The diffusion coefficient D was calculated from a linear fit on the 20 first points of the MSD curve. The slope of the fit was equivalent to 4D.

The radial (d.cos[θ]) component of a trajectory was computed by taking the angle θ between the displacement (at a given Δt) and the radial axis (Fig. S4 C). d is the norm of the displacement. The radial velocity is calculated by taking the projection of the displacement on the radial vector over the time step Δt.

For the aggregated fluorescent particles, we extracted the apparent diameter based on the intensity of the detected spot.

For estimation of centering, we defined a droplet as centered when its off-centering was <0.2, with the off-centering being the distance of the droplet centroid to the center normalized by the oocyte radius (the off-centering was equal to 1 at the cortex and to 0 at the center of the oocyte). To extract the mean centering velocity, we fitted the distance of the droplet centroid to the oocyte center as a function of time to an exponential function ($a.\exp[b.t]$). The centering velocity was then defined as $v = a.b$.

To measure the fluorescence intensity ratio of the GFP-UtrCH probe between the droplet (or the nucleus) and the cytoplasm, we drew a contour line around the droplet (or the nucleus) and then a line of the same length in the cytoplasm. We then measured the intensity along these lines and computed the ratio between the droplet (or nucleus) intensity and the cytoplasmic intensity. For Fig. 3 B, the $t$ test was performed with Prism. z-tests and Kolmogorov-Smirnov statistical tests were performed with Matlab. For the interpretation of P values, NS means there was no significant difference between the two distributions, one star means P < 0.05, two stars mean P < 0.01, three stars mean P < 0.001, and four stars mean P < 0.0001.

### Cumulative bias analysis

To check the existence of a directional bias at short time scale, we observed the displacement of object centroids at different time resolutions Δt. For each time point, we measured the angle θ between the displacement (at a given Δt) and its radial axis (Fig. S3 A, scheme). We classified the displacements in three groups: toward the center when θ lies between –30° and 30°, toward the cortex for angles >60° or under –60°, and not directional otherwise. With this classification, the displacement angle of a random trajectory had equal probability of belonging to each group (probability = 1/3). We then calculated the sum of all displacements toward the oocyte center (presenting angle –30° < θ < 30°) and compared it to the total displacement of the object (Fig. S3 A, scheme). This percentage was averaged over all displacements of one individual object and over all the different objects per condition. We referred to this score as cumulative bias.

To assess the significance of this score (i.e., the probability of obtaining a given cumulative bias only by chance), we constructed the confidence interval of the cumulative bias values for random trajectories for each time step Δt. For this, we generated 1,000 simulations of pure random (Brownian) motion of an object with size and diffusion coefficient similar to that of the tested object. We matched the simulation output intervals and total duration with the experimental ones. Using a bootstrapping technique, for each time step Δt, we extracted a given number, N, of simulations and computed their cumulative biases. We repeated this step 10,000 times, generating a distribution of cumulative bias for a random motion. The lower (upper, respectively) limit of the confidence interval was then taken as the 1% (99%, respectively) quantile. The number N of extracted simulations at each bootstrap step was taken equal to the number of experimental objects from which the cumulative bias to assess was calculated. Therefore, with this construction, a cumulative bias calculated from N objects lies within the confidence interval with a probability of 99% if their motion is Brownian. Cumulative bias outside this interval is thus likely generated by biased motion (with a P value of 0.01).

### Online supplemental material

Fig. S1 presents the outcome of simulations in the absence of any gradient of persistence of actin-positive vesicles. Fig. S2 displays the analysis of oil droplet motion at low and high temporal resolutions in prophase I. Fig. S3 measures the impact of actin microfilaments on the cumulative cortex versus center motion bias of oil droplets. Fig. S4 measures the cumulative cortex versus center motion bias of fluorescent aggregates of beads as a function of their size. Fig. S5 shows optical tweezers experiments to compare the viscous and elastic moduli of prophase I versus meiosis I oocytes. Video 1 presents an example of a simulation of a gradient of persistence of actin-positive vesicles. Video 2 presents an example of a simulation of an absence of a gradient of persistence of actin-positive vesicles. Video 3 shows an oil droplet in an oocyte maintained in prophase I observed at low temporal resolution. Video 4 shows an oil droplet in an oocyte maintained in prophase I observed at high temporal resolution. Video 5 displays an oil droplet in an oocyte maintained in prophase I treated with Cyto D and observed at high temporal resolution. Video 6 shows an oocyte injected with fluorescent beads maintained in prophase I observed at high temporal resolution. Video 7 shows an oil droplet in an oocyte undergoing meiosis I observed at low temporal resolution. Video 8 displays an oil droplet in an oocyte undergoing meiosis I and treated with Cyto D and observed at low temporal resolution. Table S1 displays the parameters used for 3D simulations.

### Acknowledgments

A. Colin is supported by a Ministère de la Recherche doctoral fellowship. G. Letort is supported by the Agence Nationale de la Recherche (ANR-16-CE13 to M.-E. Terret). This work was supported by the Agence Nationale de la Recherche (ANR-DIVACEN to M.-H. Verlhac and R. Basto; Curie Institute, number 14-CE11), by a Fondation pour la Recherche Médicale Label (DEQ20150331758 to M.-H. Verlhac), by a Paris Sciences et Lettres Aux Frontière des Labex grant (MYOOCYTE, M.-H. Verlhac as coordinator), and by an Inca grant (PLBIO 2016-270-TRAN). This work has received support from the Fondation Bettencourt Schueller and support under the program Investissements d'Avenir launched by the French Government and implemented by the Agence Nationale de la Recherche, with the references ANR-10-LABX-54 MEMO LIFE and ANR-11-IDEX-0001-02 Paris Sciences et Lettres Research University. T. Betz is supported by the European Research Council (PolarizeMe, grant number CoG 771201). N.S. Gov is the incumbent of the Lee and William Abramowitz Professorial Chair of Biophysics and acknowledges support from the Israel Science Foundation (grant number 580/12).

The authors declare no competing financial interests.

Author Contributions: R. Voituriez, Z. Gueroui, and M.-H. Verlhac supervised the work. M.-H. Verlhac did all the

microinjections and imaging of mouse oocytes. A. Colin analyzed all experimental microinjection data. G. Letort did all the 3D simulations. N. Razin and N.S. Gov did the modeling. M. Almonacid and W. Ahmed performed the optical tweezers experiments under T. Betz's supervision. W. Ahmed, T. Betz, N.S. Gov, and M.-E. Terret participated in the discussions on data interpretations together with A. Colin, G. Letort, R. Voituriez, Z. Gueroui, and M.-H. Verlhac. A. Colin, G. Letort, and M.-H. Verlhac wrote the manuscript, which was seen and corrected by all authors.

Submitted: 24 August 2019

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

## Supplemental material

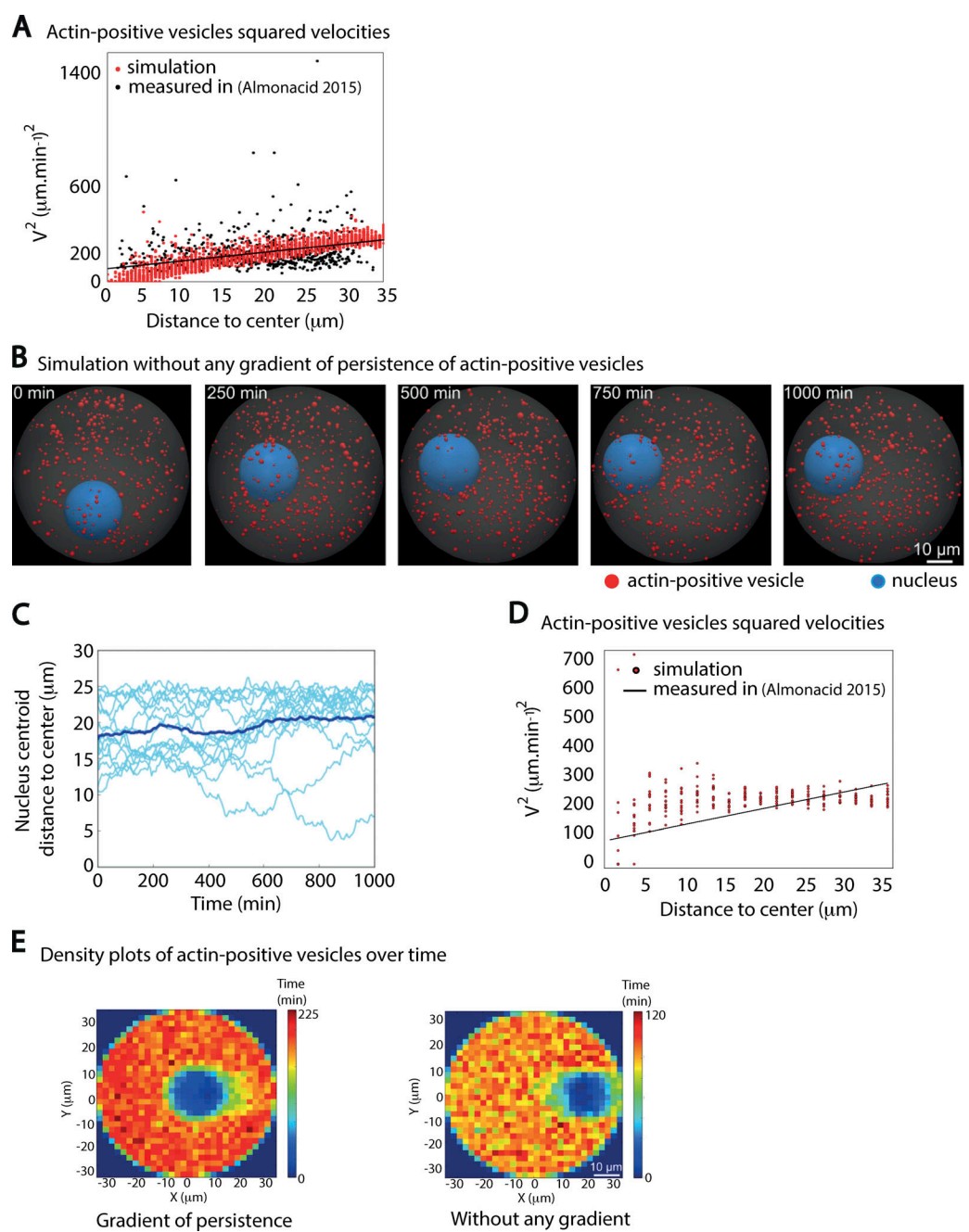

**Figure S1.** **Testing the outcomes of simulations with or without any gradient of activity. (A)** Squared velocities of actin-positive vesicles as a function of the distance to the oocyte center during the centering phase. One red dot corresponds to the averaged square velocities of vesicles in the given distance bin in one simulation, taken between 30 and 60 min of simulation only (simulations done as in Fig. 1). The black dots correspond to the experimental data from Almonacid et al. (2015) presented in Fig. 1 B, with the black line representing the fit extracted from these experimental data. **(B)** Time frames from one 3D simulation of an absence of gradient of persistence of actin-positive vesicles. The nucleus is not centered in these conditions. The nucleus is in blue, and the actin-positive vesicles are in red; the oocyte cytoplasm appears in gray. **(C)** Evolution of the distance between the nucleus centroid to the center of the oocyte as a function of time for 15 different simulations in the absence of a persistence gradient. Light blue curves correspond to individual simulations, while the dark blue curve corresponds to the mean curve for all simulations. **(D)** Squared velocity of actin-positive vesicles as a function of the distance to the oocyte center from simulations of a lack of gradient or from experimental data during the centering phase. The squared velocities of actin-positive vesicles obtained from simulations are in red, while the experimental data are in black. **(E)** Outcome from the two types of simulations on the final distribution of actin-positive vesicles. The density of actin-positive vesicles is presented as a heat map reflecting the number of vesicles per square unit during 1,000 min for the two types of simulations (with or without a gradient of persistence). To build the heat map, we aligned the simulations so that the nucleus is always on the right along the x-axis. One square unit corresponds to a square of 2.3 × 2.3 μm² in size. Left panel: Persistence gradient; right panel: without the gradient.

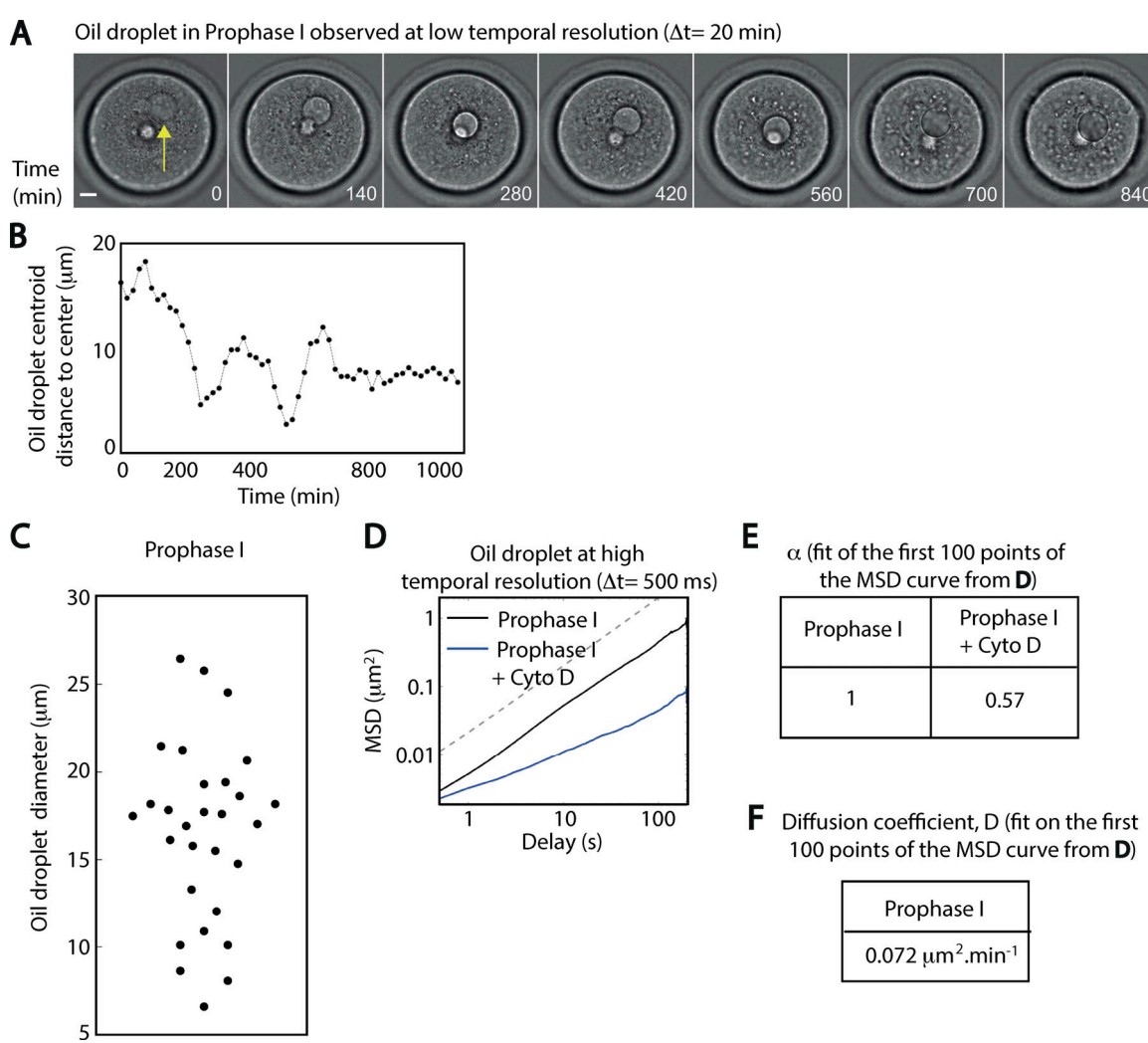

Figure S2. **Example of an oscillatory motion of a droplet around the oocyte center. (A)** Images in transmitted light of a droplet centered in prophase I. One image is shown every 140 min. Scale bar is 10 μm. The yellow arrow points at the oil droplet. **(B)** Distance of the droplet centroid to the oocyte center as a function of time (in minutes) from the images shown in A. Two oscillations of the droplet can be observed at 200 min and at 600 min. It is worth noting that the droplet encounters the nucleus when it reaches the center. **(C)** Distribution of diameter lengths for all droplets that are not central at the beginning of the experiment (*n* = 29 oocytes); four independent experiments. **(D)** MSD of oil droplets as a function of the delay (in seconds) on a log/log scale. *n* = 29 oocytes in prophase I (black curve), and *n* = 14 oocytes in prophase I + Cyto D (blue curve); three independent experiments. The dotted gray line represents a linear regression with a slope of 1. **(E)** Results obtained from the fitting of the MSD curve from D on a log-log scale. α represents the slope of the curve. The fit is done on the first 100 points of the curve. **(F)** Evaluation of the diffusion coefficient from the fit of the MSD curve from D. The fit is done on the first 100 points of the curve. The diffusion coefficient is then evaluated considering the following formula: MSD = 4.D.t (with t as the delay).

**A**

Track of oil droplet centroid

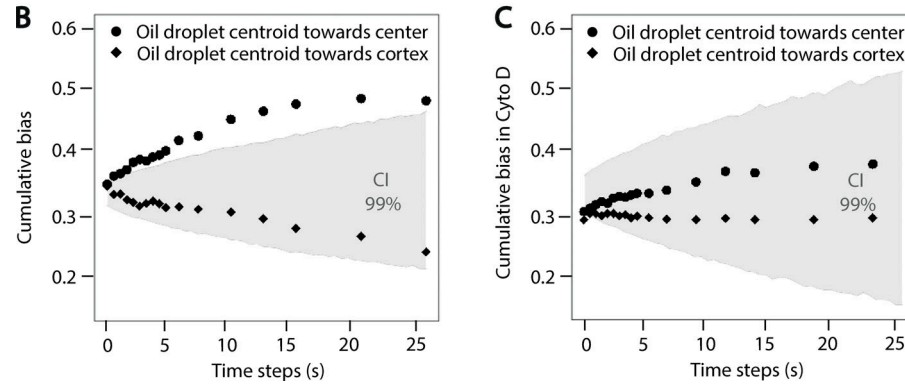

Cumulative bias of oil droplet towards:
- center (displacement for -30° < θ < + 30°) = (d1+ d4)/(d1+d2+d3+d4)
- cortex (displacement for 60° < θ < -60°) = (d2+ d3)/(d1+d2+d3+d4)

**B**

- ● Oil droplet centroid towards center
- ◆ Oil droplet centroid towards cortex

*Cumulative bias* (y-axis, 0.2 to 0.6)
*Time steps (s)* (x-axis, 0 to 25)

CI 99%

**C**

- ● Oil droplet centroid towards center
- ◆ Oil droplet centroid towards cortex

*Cumulative bias in Cyto D* (y-axis, 0.2 to 0.6)
*Time steps (s)* (x-axis, 0 to 25)

CI 99%

Figure S3.   **F-actin is responsible for the biased motion of oil droplets toward the oocyte center. (A)** Scheme explaining how the cumulative bias toward the oocyte center or cortex is measured. The black dots correspond to the centroid of objects. One track of an oil droplet centroid is presented with the directionality of motion depicted by an arrow for each time step. The nucleus is in blue. **(B)** Cumulative bias toward center and toward cortex for oil droplets. Black dots represent the oil droplet cumulative bias toward the center calculated for 17 different time steps from 29 different trajectories. Black diamonds represent the cumulative bias toward the cortex calculated for 17 different time steps from 29 different trajectories. The gray area represents the numerically constructed 99% confidence interval (CI) of cumulative bias values for *n* = 29 objects with a random motion (see Materials and methods). The cumulative biases outside this interval are likely generated by a biased motion (P value of 0.01). **(C)** Cumulative bias toward center and toward cortex for oil droplets in oocytes treated with Cyto D. Black dots represent the oil droplet cumulative bias toward the center calculated for 17 different time steps from 29 different trajectories. Black diamonds represent the cumulative bias toward the cortex calculated for 17 different time steps from 29 different trajectories. The gray area represents the numerically constructed 99% confidence interval of cumulative bias values for *n* = 29 objects with a random motion (see Materials and methods). The cumulative biases outside this interval are likely generated by a biased motion (P value of 0.01).

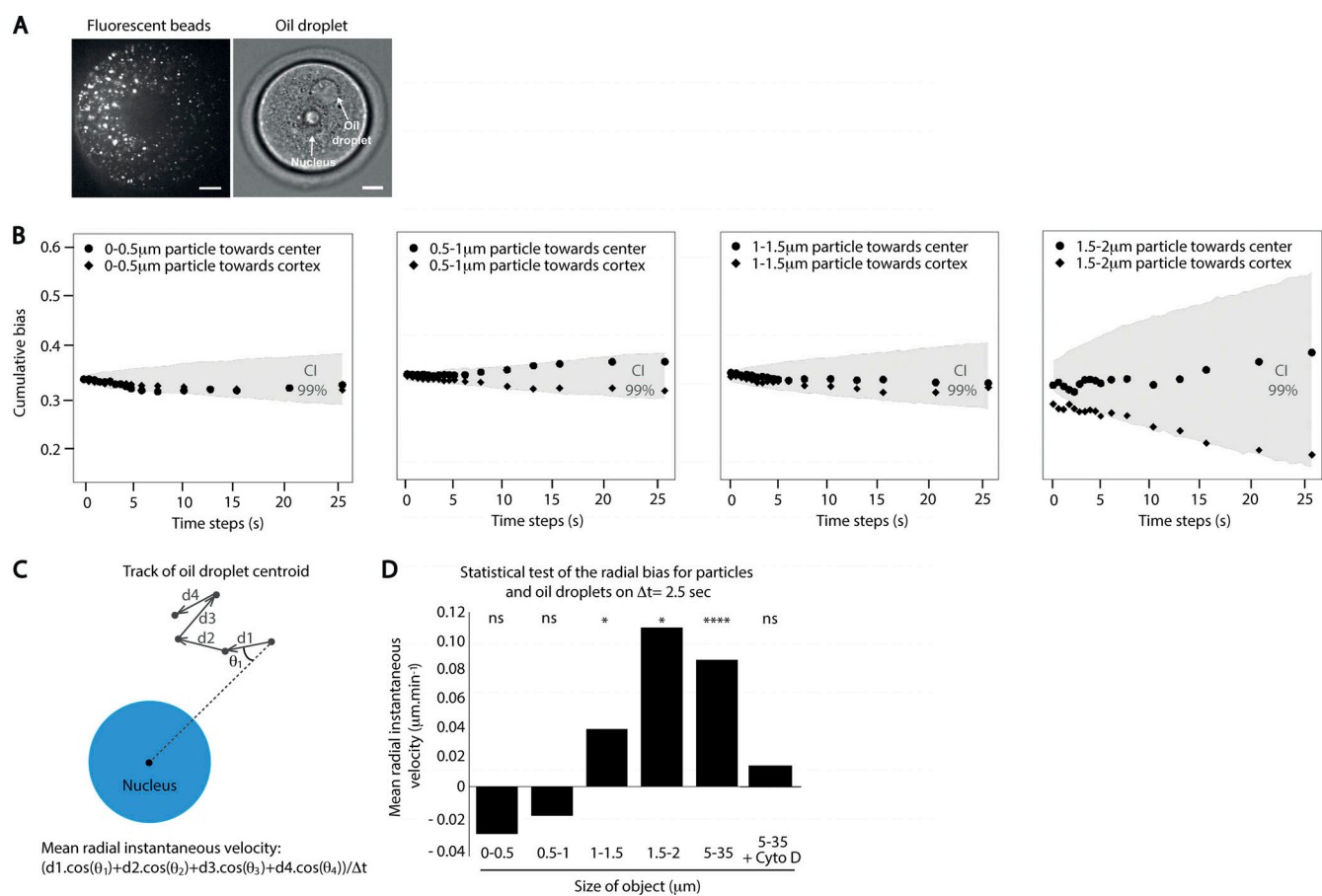

**Figure S4. Objects larger than a few micrometers are biased in their diffusion. (A)** Images of fluorescent aggregates of beads (left panel) and an oil droplet in prophase I oocytes (right panel). Aggregates of beads correspond to objects with diameters between 100 nm and 2 µm. Oil droplets correspond to objects between 5 and 30 µm in diameter. The oocyte nucleus is 25 µm wide. Scale bars are 10 µm. **(B)** Cumulative bias of the particle centroid motion for particles of varying diameters. Black dots represent the particle cumulative bias toward the center and black diamonds toward the cortex. For 0–0.5-µm particles, it was calculated from $n$ = 185 trajectories. For 0.5–1-µm particles, it was calculated from $n$ = 445 trajectories. For 1–1.5-µm particles, it was calculated from $n$ = 112 trajectories. For 1.5–2-µm particles, it was calculated from only $n$ = 13 trajectories. The gray area on all panels represents the numerically constructed 99% confidence interval (CI) of cumulative bias values for the same number and same size of object as the experimental one and presenting a random motion (see Materials and methods). The cumulative bias outside these intervals is likely generated by directionally biased motion (P value of 0.01). **(C)** Scheme explaining how the mean radial instantaneous velocity of object centroid is measured. The black dots correspond to the centroid of objects. One track of an oil droplet centroid is presented with the directionality of motion depicted by an arrow for each time step. The nucleus is in blue. **(D)** Mean radial instantaneous velocity as a function of object size. The velocity is computed from the distribution of d.cos(θ) at 5Δt (2.5 s). The P value is the probability that the distribution is significantly different from a normal distribution with the same standard deviation and centered at 0 (result of a z-test). For objects <1 µm, there is no bias toward the center. For objects >1 µm, there is a significant bias toward the oocyte center. $n$ = 5 oocytes for aggregates of particles, and $n$ = 29 oocytes for oil droplets. As presented on the figure (from left to right), the P values are: 0.17 (ns); 0.08 (ns); 0.039 (*); 0.032 (*); 4.9 × 10$^{-11}$ (****), and 0.1 (ns). ns, not significant.

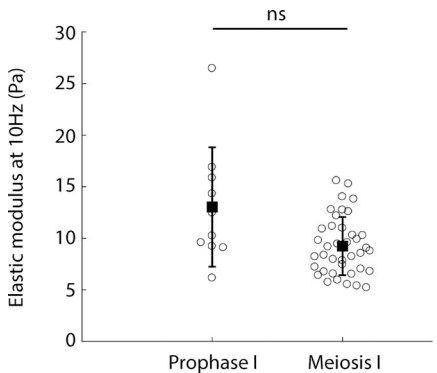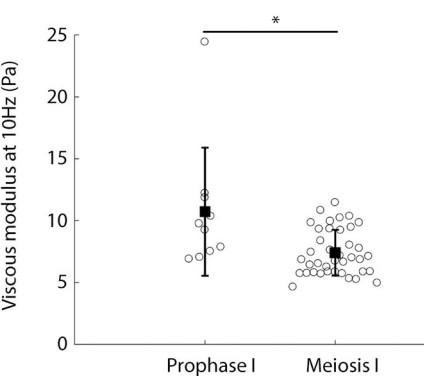

Figure S5. **Optical tweezers measurements.** Quantification of elastic and viscous moduli at 10 Hz for oocytes maintained in prophase I and oocytes undergoing meiosis I. For oocytes undergoing meiosis I, the measurements were taken at NEBD + 6 h. P values were calculated with a Kolmogorov-Smirnov statistical test (P value = 0.07 for G', the elastic modulus on the left panel, and P value = 0.02 for G'', the viscous modulus on the right panel). $n$ = 10 vesicles in seven oocytes in prophase I and $n$ = 39 vesicles in 14 oocytes in meiosis I. Mean and SEM are superimposed on the raw data; three independent experiments. ns, not significant.

Video 1. **Movie from a simulation of a gradient of persistence of actin-positive vesicles.** Time-lapse movie of an object (nucleus, blue) pushed by actin-positive vesicles (red) following a gradient of persistence. Frames are registered every 5 min. Movie duration is 1,000 min.

Video 2. **Movie from a simulation of an absence of gradient of actin-positive vesicles.** Time-lapse movie of an object (nucleus, blue) pushed by actin-positive vesicles (red) without a gradient. Frames are registered every 5 min. Movie duration is 1,000 min.

Video 3. **Oil droplet in an oocyte maintained in prophase I observed at low temporal resolution with a Δt of 20 min.** Time-lapse movie of an oocyte injected with an oil droplet in prophase I. Frames are taken every 20 min. Movie duration is 16 h.

Video 4. **Oil droplet in an oocyte maintained in prophase I observed at high temporal resolution with a Δt of 500 ms.** Time-lapse movie of an oocyte injected with an oil droplet in prophase I. Frames are taken every 500 ms. Movie duration is 400 s.

Video 5. **Oil droplet in an oocyte maintained in prophase I treated with Cyto D and observed at high temporal resolution with a Δt of 500 ms.** Time-lapse movie of an oocyte injected with an oil droplet in prophase I and treated with Cyto D. Frames are taken every 500 ms. Movie duration is 300 s.

Video 6. **Oocyte injected with fluorescent beads maintained in prophase I observed at high temporal resolution with a Δt of 500 ms.** Time-lapse movie of an oocyte injected with fluorescent beads in prophase I. Frames are taken every 500 ms. Movie duration is 500 s.

Video 7. **Oil droplet in an oocyte undergoing meiosis I observed at low temporal resolution with a Δt of 20 min.** Time-lapse movie of an oocyte injected with an oil droplet and undergoing meiosis I. Frames are taken every 20 min. Movie duration is 14 h.

Video 8. **Oil droplet in an oocyte undergoing meiosis I and treated with Cyto D and observed at low temporal resolution with a Δt of 20 min.** Time-lapse movie of an oocyte injected with an oil droplet, treated with Cyto D, and undergoing meiosis I. Frames are taken every 20 min. Movie duration is 14 h.

**Provided online is one table is Excel. Table S1 displays the parameters used for 3D simulations.**

