## [Peer Review File · The Journal of Cell Biology]

Active diffusion in oocytes non-specifically centers large objects during Prophase I and Meiosis I

Alexandra Colin, Gaele Letort, Nitzan Razin, Maria Almonacid, Wylie Ahmed, Timo Betz, Marie-Emilie TERRET, Nir Gov, Raphael Voituriez, Zoher Gueroui, and Marie-Hélène Verlhac

Corresponding Author(s): Marie-Hélène Verlhac, Collège de France and Zoher Gueroui, Ecole Normale Supérieure

Review Timeline:	Submission Date:	2019-08-24
	Editorial Decision:	2019-10-08
	Revision Received:	2019-11-14
	Editorial Decision:	2019-12-09
	Revision Received:	2019-12-12

Monitoring Editor: Alex Mogilner

Scientific Editor: Marie Anne O'Donnell

Transaction Report:

DOI: <https://doi.org/10.1083/jcb.201908195>

October 8, 2019

Re: JCB manuscript #201908195

Dr. Marie-Hélène Verlhac
Collège de France
CIRB, Collège de France 11, place Marcelin Berthelot
Paris Cedex 05 75231
France

Dear Dr. Verlhac,

Thank you for submitting your manuscript entitled "Active diffusion in oocytes non-specifically centers large objects during Prophase I and Meiosis I". The manuscript was assessed by expert reviewers, whose comments are appended to this letter. We invite you to submit a revision if you can address the reviewers' key concerns, as outlined here.

You will see that the reviewers do have some requests to address for resubmission, and several of these are for further clarification of the data presented and discussion of the potential centering mechanism.

GENERAL GUIDELINES:

Text limits: Character count for an Article is < 40,000, not including spaces. Count includes title page, abstract, introduction, results, discussion, acknowledgments, and figure legends. Count does not include materials and methods, references, tables, or supplemental legends.

Figures: Articles may have up to 10 main text figures. Figures must be prepared according to the policies outlined in our Instructions to Authors, under Data Presentation, <http://jcb.rupress.org/site/misc/ifora.xhtml>. All figures in accepted manuscripts will be screened prior to publication.

Supplemental information: There are strict limits on the allowable amount of supplemental data. Articles may have up to 5 supplemental figures. Up to 10 supplemental videos or flash animations are allowed. A summary of all supplemental material should appear at the end of the Materials and methods section.

The typical timeframe for revisions is three months; if submitted within this timeframe, novelty will

not be reassessed at the final decision. Please note that papers are generally considered through only one revision cycle, so any revised manuscript will likely be either accepted or rejected.

Thank you for this interesting contribution to Journal of Cell Biology. You can contact us at the journal office with any questions, cellbio@rockefeller.edu or call (212) 327-8588.

Sincerely,

Alex Mogilner, Ph.D.
Monitoring Editor

Marie Anne O'Donnell, Ph.D.
Scientific Editor

Journal of Cell Biology

Reviewer #1 (Comments to the Authors (Required)):

The manuscript by Colin and colleagues validates their previously proposed model for centering the nucleus in mouse oocytes using computer simulations as well as experiments. This model relies on active diffusion of cytoplasmic vesicles, which is driven by the actin cytoskeleton that is nucleated on, connects, and pulls on these vesicles. Centering results from the fact that there is a gradient in the intrinsic velocity and persistence time of vesicles from the center to the periphery, so that the 'stronger' motion at the periphery pushes any large object towards the center, where vesicles are less active.

This model is implemented in 3D computer simulations and is validated against experiments, in which the authors inject inert beads as well as oil droplets into mouse oocytes. The authors find that the model explains well the observations, whereby large objects are efficiently centered, while small particles are 'blind' to the forces generated by vesicles.

Overall, the presented data are convincing and the conclusions are well supported by observations. However, I find that the advance is relatively little as compared to the previous study by the same group that already presented the active diffusion based theory and substantial experimental validation. In addition, I would like to request that the authors clarify the following points:

1. Could the authors expand the discussion on the size dependence of forces? When compared to high resolution images of the oocyte cytoplasm, is it a valid assumption that the vesicles are the relevant elements or is it rather the mesh of actin filaments that limits the motion of larger particles but allows free diffusion of small ones?
2. Could they clarify whether they mean that the observed accumulation of actin on the surface of

nucleus and oil droplets is an active process or 'just a side effect', and whether they suspect it has a physiological function.

3. As I understand, the authors argue that active diffusion based centering is active throughout, and that spindle relocation to the cortex in meiosis I is an independent process, which dominates over the centering forces. I would like this to be stated more clearly, because the current description is rather confusing and gives the impression as if there was some direct coordination between these processes. Indeed, the active diffusion model provides no explanation for off-centering, this model can only center objects.

4. Minor point: the fit between the two lines in Fig. 1b is very poor. This needs to be explained/improved.

5. Generally, the text needs careful editing; the phrasing is often imprecise and confusingly complicated.

After addressing these points, I would in principle happy to support the publication of this manuscript in the Journal of Cell Biology - while I remain slightly concerned with the limited novelty as compared to the previous study.

Reviewer #2 (Comments to the Authors (Required)):

In this manuscript, Colin et al. analyzed the mechanism of nucleus centering in mouse oocytes. The group previously showed that the centering depends on actin and myosin Vb, and also found that actin-positive vesicle moves greater at cell periphery [Almonacid et al., 2015]. They had proposed that the gradient of a pressure pushing the nucleus is formed along the radial direction, which moves the nucleus toward the center. In the present study, the authors first conducted 3-dimensional numerical simulation and demonstrated this mechanism can position the nucleus at the center (Fig. 1). Because the mechanism relies on the pressure gradient, oil-droplet with a size comparable to the nucleus should move to the center. The authors were able to show this is actually the case (Fig. 2). The nucleus and the droplet in the cytoplasm showed similar accumulation of actin at the surface (Fig. 3) supporting their actin-dependent movements are driven by a similar mechanism. The simulation predicted the centering is dependent on the size of the object, which was consistent with the experiments using oil droplets with various sizes (Fig. 4). The authors also observed the behavior of oil droplets in Meiosis I, when the spindle moves toward off-center. The oil droplets moved toward the center even in Meiosis I (Figs. 5&6), suggesting that the mechanism of the off-centering of the spindle is unrelated to that of the nuclear centering in Prophase although both are driven by actin cytoskeleton.

This study supports the model in which nucleus centering is driven by the gradient of actin-positive vesicle activity both experimentally and quantitatively. Because nucleus centering is one of the fundamental properties of the cell, and this study progresses the understanding of it, this study is worth publication in JCB.

I would like to request the authors to address the following points before publication.

Concerns on simulation part

1. It was not clear to me what is conceptual advances from the numerical simulation. Theoretical analyses had been already conducted for 1-dimension and 2-dimension. Is conducting the calculation in 3-dimension the major progress in the present study? If so, the authors should

describe so (e.g. we confirmed that the model works in 3-D). If not, the authors should clarify novel findings made from the current 3-D simulation.

2. The values of parameters they used in 3-D simulation to recapitulate the experimental observations can be novel information. However, the values and their implications are not described or discussed in the manuscript. I request the authors to show the parameter values. It will be also nice if the authors could discuss whether the values are reasonable in the cellular context.

3. The mechanism how the driving force of nucleus centering is generated is not convincing for me. I request the authors to clarify the following points.

3-A) First, the model assumes a repulsion force between the actin-positive vesicle and the nucleus. What is the nature of the repulsion force? A simple scenario might be the collision, but the collision-based repulsion should largely depend on the physical property of the surface (e.g. softness), and it is hard for me to imagine such collision-based repulsion is working between actin-particle and nucleus. Even so, it is further unlikely that the effect is similar between the nucleus and the oil droplet.

3-B) Second, the nuclear (or oil droplet) surface has special actin-rich structure (Fig. 3) and is different from the rest of the cytoplasm. I do not think this observation is consistent with an assumption of the model, in which the actin-positive vesicles at the surface behave similarly with those in the bulk cytoplasm. Please clarify.

3-C) What is the mechanism to produce the gradient of the persistence of the movement of actin-vesicles? This might be beyond the scope of this manuscript, but the authors should at least discuss.

4. In Figure 1b, the mobility of the actin-positive vesicles in the simulation is different from that in the measurement. Why did the authors choose the simulation parameters that shows discrepancy from the experimental observation? What will happen for the nucleus centering if the authors use parameters that fit with Fig. 1b?

5. In Figure 1c, I request the authors to show experimental curve for comparison.

6. Minor point: In Fig. 1b, 1e, S1c, the authors describe the experimental results "measured in (2)". I think this should be reference #3, as ref. #2 is a review article.

Concerns on size-dependency

7. The size dependence of the object movement is interesting. Is there a statistically significant correlation in oil droplet experiment (Figs. 4c and S3c)? I could not find the number (e.g. p-value), and it was hard for me to convince myself whether there is a correlation from the plot.

8. In the size-dependency in oil droplet experiments, smaller size is informative. While the authors described that they observed oil droplets as small as 5 micro-meter in diameter in the text, there is no velocity data for those small droplets. Why is that?

9. For the smaller size objects, the authors use aggregates of the fluorescent particles. What is the physical rationale that we can consider the particle aggregate behave as a sphere that follows the Stokes' law?

10. To explain the threshold between the bias- and diffusion-dominated size, the authors discussed the Peclet number, which is interesting. What are the Peclet numbers for large and small

objects in the simulation (Fig. 4) and how are the numbers influenced by the parameter values? Also, can the authors calculate the Peclet number for the smallest oil droplet and some of the aggregates of the fluorescent particles?

COLLÈGE
DE FRANCE
—1530—

Center for Interdisciplinary Research in
Biology
UMR CNRS 7241 / INSERM U1050
Marie-Hélène Verlhac
marie-helene.verlhac@college-de-france.fr
Collège de France
11 place Marcelin Berthelot F-75005
PARIS
T. +33 1 44 27 10 82

Paris, 14th November 2019

Dear Alex Mogilner,

Dear Monitoring Editor, *The Journal of Cell Biology*

Dear Marie Anne O'Donnell,

Dear Scientific Editor, *The Journal of Cell Biology*

We would like to thank you and the two reviewers for your interest in our submitted manuscript and the positive comments as well as the constructive suggestions which certainly allowed us to better describe the phenomenon at play in mammalian oocytes.

With this letter, we are submitting a revised version of our study: “**Active diffusion in oocytes non-specifically centers large objects during Prophase I and Meiosis I**” by Alexandra Colin, Gaëlle Letort, Nitzan Razin, Maria Almonacid, Wylie Ahmed, Timo Betz, Marie-Emilie Terret, Nir S Gov, Raphaël Voituriez, Zoher Gueroui, Marie-Hélène Verlhac.

Below, we addressed all points raised by the referees. In summary, we profoundly restructured the text of the manuscript based on their comments.

We hope that you now find our work suitable for publication in *The Journal of Cell Biology*.

Yours sincerely,

Marie-Hélène Verlhac, CIRB Director, Team leader

Point by point answer to the reviewers comments

1/ Reviewer #1:

The manuscript by Colin and colleagues validates their previously proposed model for centering the nucleus in mouse oocytes using computer simulations as well as experiments. This model relies on active diffusion of cytoplasmic vesicles, which is driven by the actin cytoskeleton that is nucleated on, connects, and pulls on these vesicles. Centering results from the fact that there is a gradient in the intrinsic velocity and persistence time of vesicles from the center to the periphery, so that the 'stronger' motion at the periphery pushes any large object towards the center, where vesicles are less active. This model is implemented in 3D computer simulations and is validated against experiments, in which the authors inject inert beads as well as oil droplets into mouse oocytes. The authors find that the model explains well the observations, whereby large objects are efficiently centered, while small particles are 'blind' to the forces generated by vesicles. Overall, the presented data are convincing and the conclusions are well supported by observations.

We would like to thank the reviewer for the constructive comments which helped improve our work and its interpretation.

However, I find that the advance is relatively little as compared to the previous study by the same group that already presented the active diffusion based theory and substantial experimental validation.

We should stress out that the results concerning the non-specificity of the centering mechanism was not demonstrated in our previous experimental work; on the theory side, only proofs of principle were provided and no direct quantitative comparison to the data was done. Here we provide solid quantitative evidence using 3D-simulations coupled to new experimental validation with injected exogeneous objects of the presence of a non-specific gradient of activity able to center large objects. Using 3D-simulations coupled to experiments we also provide evidence for the presence of a cut-off size for objects (below a few microns) where this non-specific gradient of activity is not being sensed. This result is quite novel, having potential implications on the distribution of organelles in this large cell, where Golgi stacks are known to be micronized (Wassarman & Josefowicz 1978; Moreno et al 2002). Importantly, we observe here that the centering mechanism is still active in Meiosis I, which is surprising given that the spindle is getting off-centered at that time. We suggest that the mechanism is still conserved but is weaker. This decrease in centering efficiency could be important to allow for the off-centering forces of the spindle to counteract this force.

1. In addition, I would like to request that the authors clarify the following points:

Could the authors expand the discussion on the size dependence of forces? When compared to high resolution images of the oocyte cytoplasm, is it a valid assumption that the vesicles are the relevant elements or is it rather the mesh of actin filaments that limits the motion of larger particles but allows free diffusion of small ones?

We have expanded the discussion on size as required by the reviewer (p14):

“We tested the influence of object size on the centering efficiency. Small objects (below a few microns) do not show a biased movement towards the oocyte center at short time scale (2.5 sec), while large objects (above a few microns) do. Interestingly this cut-off is of the same order of magnitude as the estimated meshwork size for oocytes maintained in Prophase I (Azoury et al., 2008; Schuh and Ellenberg, 2008) and could explain why particles smaller than the mesh size diffuse freely.”

However, the mesh size cannot explain the fact that larger objects are biased in their directionality of motion. In our simulations, we show that large objects above the mesh size, hence, above a few microns, are centered with a velocity that increases proportionally with their surface. The microinjection of oil droplets displayed a similar tendency. Hence the presence of the mesh does not limit large objects from being more efficiently centered than small ones.

We also considered the size dependence of forces by calculating how the Peclet number evolved as a function of the diameter of objects (see response to Reviewer 2, point 10). Large droplets have a high Peclet number, thus their motion is dominated by active diffusion driven bias, while smaller droplets have smaller Peclet numbers, close to 1, thus more influenced by isotropic diffusion.

2. Could they clarify whether they mean that the observed accumulation of actin on the surface of nucleus and oil droplets is an active process or 'just a side effect', and whether they suspect it has a physiological function.

We apologize for not being clear on this occasion but we consider this accumulation to be a side effect due to the presence of a boundary, as described in (Vignaud et al 2012). We do not suspect it to have a physiological function.

Since this point was also raised by reviewer two, we profoundly modified this section of the manuscript and we also added (p 8):

“We however do not consider that it has a physiological relevance but rather that it is a side effect due to the presence of boundary conditions, favorable to filament nucleation, both in the case of the nuclear envelope or the oil droplet surface, as described in (Vignaud et al 2012).”

3. As I understand, the authors argue that active diffusion based centering is active throughout, and that spindle relocation to the cortex in meiosis I is an independent process, which dominates over the centering forces. I would like this to be stated more clearly, because the current description is rather confusing and gives the impression as if there was some direct coordination between these processes. Indeed, the active diffusion model provides no explanation for off-centering, this model can only center objects.

We have rewritten most of the text to make this point clearer to the reader.

4. Minor point: the fit between the two lines in Fig. 1b is very poor. This needs to be explained/improved.

The reviewer is correct, we do not observe a perfect fit. Based on the two reviewers' comments, we decided to improve our simulations. Experimental evidence of a gradient of activity is very noisy (Almonacid 2015, see New Figure S1A), in particular close to the oocyte center; this renders the determination of the exact profile inaccessible in practice. We therefore chose to make the simplest hypothesis of a linear gradient of activity, and to calibrate the model parameters to have a satisfactory match to the data, away from the cell center. This analysis has been refined in the new Figure 1B.

New Figure 1 B: comparison of the distribution of simulated (red dots) versus experimental (black line) of actin-positive vesicle squared velocities as a function of the distance to the oocyte center

We also now present the fit together with the raw experimental data in Fig. S1 A, so that the readers can appreciate the noise present in the experimental data (see below).

New Figure S1 A: comparison of the distribution of simulated (red dots) versus experimental (black dots) of actin-positive vesicle squared velocities as a function of the distance to the oocyte center

5. Generally, the text needs careful editing; the phrasing is often imprecise and confusingly complicated. After addressing these points, I would in principle happy to support the publication of this manuscript in the *Journal of Cell Biology* - while I remain slightly concerned with the limited novelty as compared to the previous study.

We have rewritten most of the text to make it clearer to the reader.

2/ Reviewer #2:

In this manuscript, Colin et al. analyzed the mechanism of nucleus centering in mouse oocytes. The group previously showed that the centering depends on actin and myosin Vb, and also found that actin-positive vesicle moves greater at cell periphery [Almonacid et al., 2015]. They had proposed that the gradient of a pressure pushing the nucleus is formed along the radial direction, which moves the nucleus toward the center. In the present study, the authors first conducted 3-dimensional numerical simulation and demonstrated this mechanism can position the nucleus at the center (Fig. 1). Because the mechanism relies on the pressure gradient, oil-droplet with a size comparable to the nucleus should move to the center. The authors were able to show this is actually the case (Fig. 2). The nucleus and the droplet in the cytoplasm showed similar accumulation of actin at the surface (Fig. 3) supporting their actin-dependent movements are driven by a similar mechanism. The simulation predicted the centering is dependent on the size of the object, which was consistent with the experiments using oil droplets with various sizes (Fig. 4). The authors also observed the behavior of oil droplets in Meiosis I, when the spindle moves toward off-center. The oil droplets moved toward the center even in Meiosis I (Figs. 5&6), suggesting that the mechanism of the off-centering of the spindle is unrelated to that of the nuclear centering in Prophase although both are driven by actin cytoskeleton.

This study supports the model in which nucleus centering is driven by the gradient of actin-positive vesicle activity both experimentally and quantitatively. Because nucleus centering is one of the fundamental properties of the cell, and this study progresses the understanding of it, this study is worth publication in JCB. I would like to request the authors to address the following points before publication.

We would like to thank the reviewer for the constructive comments which helped improve our work and its interpretation.

Concerns on simulation part

1. *It was not clear to me what is conceptual advances from the numerical simulation. Theoretical analyses had been already conducted for 1-dimension and 2-dimension. Is conducting the calculation in 3-dimension the major progress in the present study? If so, the authors should describe so (e.g. we confirmed that the model works in 3-D). If not, the authors should clarify novel findings made from the current 3-D simulation.*

Numerical simulations presented here are an implementation of the previous model. It allowed to apply the general model to our precise situation and provide quantitative comparisons to data, as opposed to our earlier works. In particular, we could check that using the range of values extracted from the experimental observations, we do obtain centering in these conditions, and with comparable

dynamics. Thus, it demonstrated quantitatively the plausibility of the model. The previous model (1-d and 2-d) did not consider hard-core repulsion between the actin-positive vesicle themselves, only between vesicle and the nucleus, thus allowing them to overlap. With our simulations, not allowing vesicle overlap, we could check that this did not affect the previous conclusion and how this impacted the spatial distribution of vesicles (Fig. S1 D). The implementation in 3-dimension was key to allow such direct comparison, using realistic values without the need to convert from 2D to 3D. We were also able to directly test the effect of the variation of the number and size of vesicles during Meiosis I on the centering efficiency by using the measured values. The flexibility of our agent-based model also allowed to explore the impact of the size of object on their behavior. It permitted for the first time to give an order of magnitude of the threshold of size for centering objects.

2. The values of parameters they used in 3-D simulation to recapitulate the experimental observations can be novel information. However, the values and their implications are not described or discussed in the manuscript. I request the authors to show the parameter values. It will be also nice if the authors could discuss whether the values are reasonable in the cellular context.

Parameter values that could be directly linked to the cellular context were fixed from available in-vivo observations. The other parameters were model-related and normalized, so that their values were not informative. However, we can note that the shape of the persistence gradient that allowed to mimic the actin-positive vesicle overall velocity was such that the persistence of vesicle at the cortex was 10-fold the one at the center of the oocyte.

The reviewer makes a very good point and to answer it, we have added a Table in the manuscript (Material and Methods section) which clearly states the value of the parameters used in the simulations (Table 1, see below).

Parameter name	Value	Comment/Reference
Simulation		
Cell radius	35 μm	From (Almonacid et al., 2015; Schuh, 2011)
Time step, dt	0.001 min	Taken small to avoid numerical errors
Final time	1000 min	Long enough for nucleus to be centered (Almonacid et al., 2015)
Nucleus		
Mean radius	12.5 μm	From (Almonacid et al., 2015)
Motility	0	To test only the effect of the activity gradient
Vesicle-Nucleus repulsion, C_v	600	Empirical value, chosen for effective repulsion
Cortex-Nucleus repulsion, C_d	600	Empirical value, taken equal to C_v
Actin-Positive vesicles		
Mean radius	0.5 μm	Measured in (Almonacid et al., 2015; Holubcová et al., 2013; Schuh, 2011)
Number	500	Measured in (Almonacid et al., 2015; Holubcová et al., 2013; Schuh, 2011)
Motility	2	Instantaneous velocity of Brownian motion normalized by the viscosity of the medium. Chosen such that the mean vesicle velocity is around 13 $\mu\text{m}\cdot\text{min}^{-1}$ (Almonacid et al., 2015; Schuh, 2011)
Persistence gradient coefficient, τ_0	0.001 min	Chosen to obtain comparable velocity gradient to (Almonacid et al., 2015)
Persistence gradient slope, τ_r	0.25 μm^{-1}	Chosen to obtain comparable velocity gradient to (Almonacid et al., 2015)

Vesicle-vesicle repulsion C_r	400	Empirical value, chosen for effective repulsion
Cortex-vesicle repulsion C_d	400	Empirical value, taken equal to C_r

Table 1: parameters used in 3D-simulations

3. *The mechanism how the driving force of nucleus centering is generated is not convincing for me. I request the authors to clarify the following points.*

3-A) *First, the model assumes a repulsion force between the actin-positive vesicle and the nucleus. What is the nature of the repulsion force? A simple scenario might be the collision, but the collision-based repulsion should largely depend on the physical property of the surface (e.g. softness), and it is hard for me to imagine such collision-based repulsion is working between actin-particle and nucleus. Even so, it is further unlikely that the effect is similar between the nucleus and the oil droplet.*

The repulsion force considered here is steric repulsion (Drasdo et al 2007; Belmonte et al 2008; Berry and Chaté 2014; Camley et al 2017; Macklin et al 2012; Chow et Skolnick 2015). In other words, it means that when two objects come into contact, a sharp increasing repulsive force prevents physical overlap. For numerical efficiency, in practice, we chose a simple linear repulsion force, using the same definition as in (Macklin et al 2012; Ghaffarizadeh et al 2018; Letort et al 2018). Eventually, the active pressure effect, which works at long-range, does not strongly depend on this choice of short-range steric repulsion. For objects, there is nonetheless an active pressure, so this choice is not a crucial one in our 3D-simulations. In addition, we made the simplifying hypothesis that the direction of the propulsion force remains independent of any interaction with other particles (no torque; see Solon and et al, Nat Phys 2015).

The repulsion strength does indeed depend on the physical properties of the colliding objects. Since the precise physical properties of objects are not known, we kept for simplicity the same potential description for all objects, but varied the coefficient of the repulsive force, between vesicle-vesicle and vesicle-nucleus interactions (empirical values). This eventual difference in interaction between oil droplet/vesicles and nucleus/vesicles could explain the observed difference in centering velocities between oil droplets and nucleus.

Moreover, the objects are slowed down differently according to their size (following the Stokes's law), so the repulsive force does not have the same effect on vesicle as it has on the nucleus. Overall, in our observations and simulations, the centering mechanism properties were similar (though not exactly equal), and depended more on the size of the objects than on their deformability or specific choice of interactions.

3-B) *Second, the nuclear (or oil droplet) surface has special actin-rich structure (Fig. 3) and is different from the rest of the cytoplasm. I do not think this observation is consistent with an assumption of the model, in which the actin-positive vesicles at the surface behave similarly with those in the bulk cytoplasm. Please clarify.*

We apologize for not being clear on this occasion but we consider this accumulation to be a side effect due to the presence of a boundary more than to anything else, as described in (Vignaud et al 2012). We do not suspect it to have a physiological function.

Since this point was also raised by reviewer one, we profoundly modified this part of the manuscript and we also added (p 8):

“We however do not consider that it has a physiological relevance but rather that it is a side effect due to the presence of boundary conditions, favorable to filament nucleation, both in the case of the nuclear envelope or the oil droplet surface, as described in (Vignaud et al 2012).”

3-C) *What is the mechanism to produce the gradient of the persistence of the movement of actin-vesicles? This might be beyond the scope of this manuscript, but the authors should at least discuss.*

This is an excellent point. Unfortunately, the fundamental reason underlying the persistence of the

movement of actin-positive vesicle, observed in (Schuh 2011) and confirmed in (Almonacid et al 2015), remains unaddressed. We can only speculate on its origin. It might be related to the presence of a very dynamic equilibrium that sustains an important trafficking of Rab11a containing actin-positive vesicle from the center to the periphery (Schuh 2011) . It might be also due to a potential positive feed-back loop residing in the massive enrichment of F-actin in and extending from the cortex, and/or the accumulation of the Myosin Vb at the cortex (Schuh 2011).

We have added these speculations in the Discussion.

4. In Figure 1b, the mobility of the actin-positive vesicles in the simulation is different from that in the measurement. Why did the authors choose the simulation parameters that shows discrepancy from the experimental observation? What will happen for the nucleus centering if the authors use parameters that fit with Fig. 1b?

The reviewer is correct, we do not observe a perfect fit. Based on the two reviewers' comments, we decided to improve our simulations. Experimental evidence of a gradient of activity is very noisy (Almonacid 2015, see New Figure S1A), in particular close to the oocyte center; this renders the determination of the exact profile inaccessible in practice. We therefore chose to make the simplest hypothesis of a linear gradient of activity, and to calibrate the model parameters to have a satisfactory match to the data, away from the cell center. This analysis has been refined in the new Figure 1B.

New Figure 1 B: comparison of the distribution of simulated (red dots) versus experimental (black line) of actin-positive vesicle squared velocities as a function of the distance to the oocyte center

We also now present the fit together with the raw experimental data in Fig. S1 A, so that the readers can appreciate the noise present in the experimental data (see below).

New Figure S1 A: comparison of the distribution of simulated (red dots) versus experimental (black dots) of actin-positive vesicle squared velocities as a function of the distance to the oocyte center.

5. In Figure 1c, I request the authors to show experimental curve for comparison.

In Figure 1 C, we showed the evolution of the distance of the simulated object to the oocyte center over time. Experimental data for comparison would then be the distance of the nucleus to the oocyte center, corresponding to the experiments done in (Almonacid et al 2015). However, in these data, we measured the absolute position of the nucleus over time, from which the speed was extracted, but we did not have access to the position of the nucleus relative to the oocyte center.

By assuming that the nuclei get centered at the end of all tracks, we can extract the oocyte center position and the relative position of the nucleus. With this approximation, we can obtain the graph below, where simulations are represented in blue, corresponding to Fig. 1 C, and experimental data are added in black.

Figure Legend: Evolution of the distance of the nucleus to the **estimated** oocyte center over time. Experimental measures from (Almonacid et al 2015) are presented in black and end position corresponds to the last point for each trajectory. In simulated trajectories, in blue, the distance to the exact center of the oocyte was calculated. The bolder lines represent the average trajectories for each condition (experimental, black; simulated, blue).

Since the nucleus is fluctuating around the oocyte center (Almonacid et al 2015 and 2019), it is very unlikely that the last point of the nucleus trajectory corresponds to the exact center of the oocyte. It actually can be observed in the simulated trajectories (Fig. 1 C) where the distance of the nucleus-like object stabilizes around 3 to 4 μm away from the oocyte center. Thus, by considering the last point of the nucleus trajectory in the analysis of the experimental data as the oocyte center, we clearly underestimate the actual distance to the center. Therefore, it is not surprising that the experimental data curves lie below the simulated ones. To take this into account, we shifted the experimental data by a few microns vertically, to account for the oscillations of the nucleus around the oocyte centroid at steady state, only for the sake of visualization. Moreover, experimental observations start after the beginning of nucleus motion to the center, for obvious technical reasons (it takes time to microinject and let the cRNA express), with a nucleus closer to the center than in the simulations. Thus to compare directly the trajectories, we also added a delay in the experimental starting point, such that the beginning of the experimental curves match with the simulated ones (see Figure below). In this **artificial** configurations, we can observe that the fit between simulations and experimental observations is actually quite good (see Figure below).

Figure Legend: Evolution of the distance of the nucleus to the **estimated** oocyte center over time, with **shifted** experimental data. Experimental measures from (Almonacid et al 2015) are presented in black and center position was estimated as the last point of each trajectory. All the experimental trajectories were **translated** both vertically and horizontally to account for the imprecise centering of the nucleus and the difference in timing. In simulated trajectories, in blue, the distance to the exact center of the oocyte was calculated. The bolder lines represent the average trajectories for each condition (experimental, black; simulated, blue).

However, since this shifting is artificial and could be misleading, we prefer not to show it in the revised manuscript.

6. *Minor point: In Fig. 1b, 1e, S1c, the authors describe the experimental results "measured in (2)". I think this should be reference #3, as ref. #2 is a review article.*

The reviewer is correct and we have modified it. We apologize for this mistake.

Concerns on size-dependency

7. *The size dependence of the object movement is interesting. Is there a statistically significant correlation in oil droplet experiment (Figs. 4c and S3c)? I could not find the number (e.g. p-value), and it was hard for me to convince myself whether there is a correlation from the plot.*

Unfortunately, there is no observable correlation with our available experimental data. It could come from the low number of experimental data (15), from the differences in their initial positioning which affect their speed (larger droplet were usually closer to the center) or from the limited range of sizes that we could follow on long timescale with the oil droplet technique. As we detailed in the Material and Methods section, injecting oil droplets was actually challenging and clearly not a calibrated process: the oil has a very high surface tension (its density is twice that of water) and thus we had to manually apply the maximal pressure (7000 hPa) coming out from the Eppendorf microinjector to force it to go out of the pipet tip into the oocyte cytoplasm. Hence droplets of different sizes were produced only by visual check after microinjection. Furthermore, as explained below (answer to point 8), it was impossible to follow by transmitted light droplets smaller than 5 µm.

We have modified the text as follows (p. 9-10) and removed the misleading experimental fit in Fig. 4 C:

“While the outcome of the simulations indicated a clear correlation between object size and centering efficiency (Fig. 4 C, multicolor points fitted with a linear regression depicted as a blue dotted line), our experimental points coming from oil droplets, that we could follow throughout the whole movie duration up to their centering, with diameters between 8 to 26 µm, showed only a modest trend (Fig.

4 C, black triangles). This could come from the low number of experimental data (15) or from the limited range of sizes that we could follow on long timescale with the oil droplet technique. Nevertheless, more efficient centering observed for larger object is expected from the Archimedes-like property of the centering force (Razin et al., 2017a). Interestingly, it was observed previously for multi-cellular clusters in a different model system, the *Drosophila* embryos (Cai et al., 2016)."

8. In the size-dependency in oil droplet experiments, smaller size is informative. While the authors described that they observed oil droplets as small as 5 micro-meter in diameter in the text, there is no velocity data for those small droplets. Why is that?

Unfortunately, there is a technical difficulty since oil droplets smaller than 5 μm are not refringent enough by transmitted light to be followed in the crowd environment of the mouse oocyte cytoplasm.

9. For the smaller size objects, the authors use aggregates of the fluorescent particles. What is the physical rationale that we can consider the particle aggregate behave as a sphere that follows the Stokes' law?

The reviewer makes a valid point, the aggregates of a few beads are not spherical (see Supplementary Video S6 added in the revised manuscript for more clarity on this point). However, starting at a few radii, the flow fields of spherical versus non-spherical objects should become indistinguishable.

10. To explain the threshold between the bias- and diffusion-dominated size, the authors discussed the Peclet number, which is interesting. What are the Peclet numbers for large and small objects in the simulation (Fig. 4) and how are the numbers influenced by the parameter values? Also, can the authors calculate the Peclet number for the smallest oil droplet and some of the aggregates of the fluorescent particles?

The reviewer is correct and we have amended the Discussion part on the Peclet number as follows: "The Peclet number, P , is the ratio between the time for diffusion across the system to the time for directed (biased) advection: $P = \tau_D / \tau_A$. The diffusion time is given by $\tau_D = L^2 / D$, where L is the system size and $D = D_T + D_A$ is the diffusion coefficient, that arises from both thermal (T) and active forces (A). The advection time, due to the net forces acting on an object, is given by $\tau_A = L / v$, where in our system the net advection velocity of the object, v , is given by the unbalanced collisions with the actin-positive vesicles that display an activity gradient.

The Peclet number is therefore: $P = \frac{\tau_D}{\tau_A} = \frac{Lv}{D}$. Large Peclet number means that the diffusion time is long compared to the advection time, so that the advection process is dominant. Small Peclet number corresponds to diffusion-dominated dynamics.

Assuming that despite the contribution of active effects the diffusion coefficient satisfies the equilibrium scaling as such $D \sim \frac{1}{r}$, and that the advection velocity scales as $v \sim r^2$, as was described in (Cai et al., 2016), the Peclet number is eventually found to scale as: $P \sim r^3$; it is thus large for large objects and small for small objects."

Also, to fully address the reviewer's comments, we first calculated the Peclet number in our simulations. For this, we simulated short-time trajectories of objects of radius 1, 6, 12.5 and 18 μm to obtain the diffusion coefficient of the object for each size. Then, taking the diameter of the object $2r$ as the characteristic length L_c and $v=0.00024 r^2$ (from the fit calculated in Fig. 4 C), and using the above formula, we obtained Peclet numbers of 0.05 ($r = 1 \mu\text{m}$), 4 ($r = 6 \mu\text{m}$), 30 ($r = 12.5 \mu\text{m}$) and 140 ($r = 18 \mu\text{m}$) respectively. This confirmed that for small objects, the Peclet number was below 1, so that thermal diffusion dominated advection (pressure from the gradient here).

Then, in the experiments, we did not have access to the centering velocity of the small aggregate as we could only follow them on short time scales. Thus, their Peclet number could not be calculated. For the oil droplet, the diffusion coefficient ($0.07 \mu\text{m}^2 \cdot \text{min}^{-1}$) was calculated from the pooled trajectories of all droplet sizes. Using this approximate value, we calculated the Peclet number as a function of oil droplet diameter (see Figure below).

Figure Legend: Calculated Peclet number as a function of oil droplet diameter

The highest Peclet number was 128, for an oil droplet of $8.5 \mu\text{m}$ radius and the smallest was 1.3 for an oil droplet of $5 \mu\text{m}$ radius. The Peclet number of oil droplets were above 1, consistent with the observed centering (active diffusion dominated), but close to 1 for small droplets (under $6 \mu\text{m}$ radius), suggesting a shift towards a thermal diffusion dominated regime.

We are sharing these technical elements on the measure of the Peclet numbers with the reviewer, but for the sake of clarity of the text, we presented a short version in the Discussion (p. 15):

“In agreement with the theory, the Peclet number in the simulations of object of a diameter of $2 \mu\text{m}$ (respectively of 6 , 12.5 and $18 \mu\text{m}$ diameter) was 0.05 (respectively 4 , 30 and 140). This confirmed that for small objects, the Peclet number was below 1, so that thermal diffusion dominated advection, while the motion of larger objects was dominated by active diffusion. In the experiments, oil droplets presenting a diameter between 8 and $10 \mu\text{m}$ ($n=3$) had Peclet numbers between 1 and 2, whereas oil droplets of a diameter larger than $10.5 \mu\text{m}$ ($n=12$) had Peclet numbers between 4 and 128. All Peclet numbers for oil droplets were above 1, indicative of a regime dominated by active diffusion, consistent with the observed centering of oil droplets. Interestingly, small droplets (of a diameter below $10 \mu\text{m}$) presented Peclet numbers close to 1, suggesting an increasing contribution of thermal diffusion compared to active diffusion on the object motion.”

December 9, 2019

RE: JCB Manuscript #201908195R

Dr. Marie-Hélène Verlhac
Collège de France
CIRB, Collège de France 11, place Marcelin Berthelot
Paris Cedex 05 75231
France

Dear Dr. Verlhac:

Thank you for submitting your revised manuscript entitled "Active diffusion in oocytes non-specifically centers large objects during Prophase I and Meiosis I". We would be happy to publish your paper in JCB pending final revisions necessary to meet our formatting guidelines (see details below).

- Suggested edits to the abstract for length or clarity:

"Nucleus centering in mouse oocytes results from a gradient of actin-positive vesicle activity and is essential for developmental success. Here, we analyze 3D model simulations to demonstrate how a gradient in the persistence of actin-positive vesicles can center objects of different sizes. We test model predictions by tracking the transport of exogenous passive tracers. The gradient of activity induces a centering force, akin to an effective pressure gradient, leading to the centering of oil droplets with velocities comparable to nuclear ones. Simulations and experimental measurements show that passive particles subjected to the gradient exhibit biased diffusion towards the center. Strikingly, we observe that the centering mechanism is maintained in Meiosis I despite chromosome movement in the opposite direction, thus it can counteract a process that specifically off-centers the spindle. In conclusion, our findings reconcile how common molecular players can participate in the two opposing functions of nucleus centering versus off-centering."

- Provide main and supplementary text as separate, editable .doc or .docx files

- Provide figures as separate, editable files according to the instructions for authors on JCB's website, paying particular attention to the guidelines for preparing images at sufficient resolution for screening and production

- Provide tables as excel files

- Add a paragraph after the Materials and Methods section briefly summarizing the online supplementary materials

A. MANUSCRIPT ORGANIZATION AND FORMATTING:

Full guidelines are available on our Instructions for Authors page, <http://jcb.rupress.org/submission-guidelines#revised>. **Submission of a paper that does not conform to JCB guidelines will delay the acceptance of your manuscript.**

B. FINAL FILES:

-- High-resolution figure and video files: See our detailed guidelines for preparing your production-ready images, <http://jcb.rupress.org/fig-vid-guidelines>.

Thank you for this interesting contribution, we look forward to publishing your paper in Journal of Cell Biology.

Sincerely,

Alex Mogilner, Ph.D.
Monitoring Editor

Marie Anne O'Donnell, Ph.D.
Scientific Editor

Journal of Cell Biology

Reviewer #2 (Comments to the Authors (Required)):

The authors responded carefully to my concerns, and the responses were satisfactory to me. The conceptual advances from the manuscript may not be large compared to the authors' previous work, but I value this work as an important step forward for further study on this interesting subject.

I thus support the publication in JCB.

A very minor comment is that the authors use "." (period) as the multiplication sign (and "-" as well in another case), and this was a little confusing for me.